# T cell infiltration into the brain triggers pulmonary dysfunction in murine Cryptococcus-associated IRIS

Tasuku Kawano [1,2,8], Jinyan Zhou[1,3,8], Shehata Anwar[1,4,8], Haneen Salah[1,5], Andrea H. Dayal [1,5], Yuzuki Ishikawa [1,5], Katelyn Boetel [1,5], Tomoko Takahashi [2], Kamal Sharma[6] & Makoto Inoue [1,3,7] ✉

*Cryptococcus*-associated immune reconstitution inflammatory syndrome (C-IRIS) is a condition frequently occurring in immunocompromised patients receiving antiretroviral therapy. C-IRIS patients exhibit many critical symptoms, including pulmonary distress, potentially complicating the progression and recovery from this condition. Here, utilizing our previously established mouse model of unmasking C-IRIS (CnH99 preinfection and adoptive transfer of CD4$^+$ T cells), we demonstrated that pulmonary dysfunction associated with the C-IRIS condition in mice could be attributed to the infiltration of CD4$^+$ T cells into the brain via the CCL8-CCR5 axis, which triggers the nucleus tractus solitarius (NTS) neuronal damage and neuronal disconnection via upregulated ephrin B3 and semaphorin 6B in CD4$^+$ T cells. Our findings provide unique insight into the mechanism behind pulmonary dysfunction in C-IRIS and nominate potential therapeutic targets for treatment.

*Cryptococcus*-associated immune reconstitution inflammatory syndrome (C-IRIS) is a condition that occurs when a patient infected with *Cryptococcus neoformans* (Cn) has an overactive immune system during reconstitution, endangering their life[1]. C-IRIS is a common manifestation of cryptococcal meningitis (CM) and is characterized by central nervous system (CNS) complications[1]. Clinical presentations of C-IRIS include headache, fever, cranial neuropathy, alteration of consciousness, lethargy, memory loss, meningeal irritation signs, and visual disturbance[2]. In addition, non-neurological presentation, including lymphadenopathy and pulmonary disease, can occur in C-IRIS patients[3–8]. These symptoms typically have a subacute onset, although acute and chronic onsets can also be observed[9]. Cn infections and CM are common in human immunodeficiency virus (HIV) patients receiving antiretroviral therapy (ART), whose immune systems,

particularly T cells, undergo reconstitution[10]. Thus, the incidence of C-IRIS is frequently reported. For instance, approximately 25–30% of HIV patients with Cn infections develop IRIS within the first four months of ART, with an average mortality rate of 8–30%[11]. Therefore, understanding the mechanisms by which an overactive immune system triggers life-threatening CNS damage is essential for preventing lethal outcomes in C-IRIS patients.

A retrospective cohort study shows that pulmonary disease occurs in approximately 20% of C-IRIS cases[8], such that patients may present with brain/uncal herniation[12], pulmonary nodules[3,8], and pleural effusions, which are rarely encountered[8]. Despite the reports, little is understood about its etiology and pathogenesis, making clinical diagnosis and treatment highly inefficient. On top of this, few have investigated the involvement of CNS and the

[1]Department of Comparative Biosciences, The University of Illinois at Urbana-Champaign, 2001 South Lincoln Avenue, Urbana, IL 61802, USA. [2]Division of Pathophysiology, Department of Pharmaceutical Sciences, Faculty of Pharmaceutical Sciences, Tohoku Medical and Pharmaceutical University, 4-4-1 Komatsushima Aoba-Ku, Sendai, Miyagi 981-8558, Japan. [3]Neuroscience Program, The University of Illinois at Urbana-Champaign, 405 North Matthews Avenue, Urbana, IL 61801, USA. [4]Department of Pathology, Faculty of Veterinary Medicine, Beni-Suef University (BSU), Beni-Suef 62511, Egypt. [5]School of Molecular and Cell Biology, The University of Illinois at Urbana-Champaign, 407 South Goodwin Avenue, Urbana, IL 61801, USA. [6]Department of Anatomy and Cell Biology, University of Illinois, Chicago, 808 S. Wood Street, Chicago, IL 60612, USA. [7]Beckman Institute for Advanced Science and Technology, 405 North Matthews Avenue, Urbana, IL 61801, USA. [8]These authors contributed equally: Tasuku Kawano, Jinyan Zhou, Shehata Anwar. ✉e-mail: makotoi@illinois.edu

mechanism of pulmonary dysfunction. Thus, it is imperative to elucidate the mechanistic cascade of pulmonary pathology in the progression of C-IRIS.

Pathology of C-IRIS is characterized by both neurological and non-neurological lesions. Recently, we have developed a mouse model of unmasking C-IRIS using immunocompromised $Rag1^{-/-}$ mice, which lack T and B cells, with intranasal (i.n.) infection of Cn serotype A H99 (CnH99, 100 yeasts) and intravenous (i.v.) transfer of CD4[+] T cells ($10^6$ cells) three weeks after CnH99 infection (hereinafter C-IRIS condition/mice)[13]. Transfer of CD4[+] T cells into $Rag1^{-/-}$ mice acted as reconstitution. This mouse model showed manifestations of weight loss, high mortality, systemic upregulation of pro-inflammatory cytokines, elevated levels of CD4[+] T cells in the lungs, infiltration of CD4[+] T cells into the brain, and brain edema[13]. On the other hand, $Rag1^{-/-}$ mice with CnH99 infection alone or CD4[+] T cell transfer alone did not show these phenomena[13]. Notably, C-IRIS murine model showed a high amount of T helper type 1 cells (Th1) infiltration into the brain and Th1-mediated symptoms. Importantly, no significant histopathological lesions were observed in the lung tissues, while cerebral edema and vacuolization features were observed in the hindbrains of C-IRIS mice[13].

The nucleus tractus solitarius (NTS), located in the hindbrain, is formed by diverse groups of neurons in the dorsolateral medulla[14]. Previous research has shown that the NTS is the first synaptic station of cardiorespiratory afferent inputs, and plays a pivotal role in processing and relaying visceral afferent information to other nuclei in the brainstem, forebrain, and spinal cord[14]. Bilateral NTS lesions can lead to neurogenic respiratory failure[15]. Moreover, cerebrospinal fluid (CSF) analyses from patients revealed higher proportions of CCR5[+] T cells[16] and elevated levels of several chemokines, such as CCL3[17], a CCR5 ligand, during C-IRIS, which may be predictive of and associated with the development of C-IRIS. In light of these reported evidence, we hypothesize that the rapidly reconstituted and infiltrated T cells into the brain in a chemokine fashion may govern the cellular function of the NTS neurons, and such abnormalities may be linked to respiratory dysfunction in C-IRIS, prompting further investigation and examination.

In the present study, utilizing our previously established mouse model, we investigated the mechanisms of pulmonary dysfunction in C-IRIS-induced $Tcra^{-/-}$ mice. $Tcra^{-/-}$ mice (lack T cells only) were used here to align with the clinical low baseline CD4[+] T cell number and intact B cell number before a rapid immune reconstitution, such as ART. We demonstrated that pulmonary dysfunction under the C-IRIS condition in mice is due to respiration-controlling NTS neuron damage induced by the upregulated ephrin B3 and semaphorin 6B, neurite retraction molecules, expressed on the infiltrated CD4[+] T cells. Moreover, we investigated and demonstrated that the CCL8-CCR5 axis is crucial for CD4[+] T cell infiltration into the brain under the C-IRIS condition. Our findings provide unique insight into the mechanism behind pulmonary dysfunction in C-IRIS and nominate potential therapeutic targets for treatment.

## Results
### Pulmonary dysfunction and mortality under the C-IRIS condition
Because pulmonary manifestations have been reported in C-IRIS patients[3,8,12], we examined if our previously established C-IRIS model[13] (CnH99 preinfection for three weeks followed by CD4[+] T cell transfer into $Rag1^{-/-}$ mice) would show any pulmonary dysfunctions. At five days after CD4[+] T cell transfer, respiratory rate was significantly reduced in the C-IRIS group compared with the three control cohorts (naive $Rag1^{-/-}$ mice, $Rag1^{-/-}$ mice with CnH99 infection alone, and $Rag1^{-/-}$ mice with CD4[+] T cell transfer alone) (Supplementary Fig. 1A). Because HIV primarily impairs T cell functions and patients with C-IRIS have low baseline CD4[+] T cells and intact B

cell numbers prior to ART, we also examined pulmonary functions and mortality in $Tcra^{-/-}$ mice (Fig. 1A), which lack T cells but retain B cells. When $Tcra^{-/-}$ mice were preinfected with CnH99 (100 yeasts, i.n.) and transferred CD4[+] T cells ($10^6$ cells, i.v.) (C-IRIS condition, detailed in Fig. 1A), the respiratory rate and oxygen saturation levels were significantly reduced compared with control groups (naive $Tcra^{-/-}$, $Tcra^{-/-}$ mice with CnH99 infection alone, and $Tcra^{-/-}$ mice with CD4[+] T transfer alone) (Fig. 1B). Note: baseline oxygen saturation in naive mice in our study is lower than the normal 95–98% range in young C57BL6 mice. That may be due to isoflurane anesthetized condition[18] and using 16–20-week-old $Tcra^{-/-}$ mice[19]. On the other hand, $Tcra^{-/-}$ and $Rag1^{-/-}$ mice with the less virulent serotype D of Cn strain 52D (Cn52D, 100 yeasts) preinfection and CD4[+] T cell transfer had no mortality and no pulmonary dysfunction (Supplementary Fig. 1B–D). We also conducted whole-body plethysmography (WBP), where mice were in an unanesthetized and unrestrained state, and their pulmonary functions were recorded. Compared to the three control groups, C-IRIS mice were found to have lower breaths per minute (BPM), increased total time (TT) per cycle, and increased expiration time (ET), while no significant difference was observed in inspiration time (IT), peak inspiratory flow (PIF), peak expiratory flow (PEF), and minute volume (MV) (Fig. 1C). Survival analysis was also conducted in $Tcra^{-/-}$ mice with C-IRIS, in which we found significantly higher mortality rates than the control groups (Fig. 1D), similar to our previous results in $Rag1^{-/-}$ mice[13]. These results identify pulmonary dysfunction in our C-IRIS mouse model and suggest that pulmonary dysfunction under the C-IRIS condition could contribute to mortality. Therefore, it is essential to identify the mechanisms of pulmonary dysfunction under the C-IRIS condition.

### CD4[+] T cells in the brain trigger pulmonary dysfunction under the C-IRIS condition
We previously have shown that a minor (non-significant) difference was observed in lung histology in three control cohorts and C-IRIS $Rag1^{-/-}$ mice[13], suggesting that histopathological lung damage is not involved in pulmonary dysfunction. However, C-IRIS $Rag1^{-/-}$ mice showed brain damage and CnH99 and CD4[+] T cell infiltration into the brain[13]. Here we also found that C-IRIS $Tcra^{-/-}$ mice showed significant brain structure abnormalities and exhibited a soap bubble appearance in the cleared whole brain using Ultramicroscopy, indicating multicystic encephalomalacia (Fig. 2A and Supplementary Fig. 2A). Cryptococcal cells have been shown to colonize brain tissues[20]; using mCherry fluorescent CnH99[21], we also observed colonization of cryptococcus near the tissue damage sites in the brains of C-IRIS mice (Fig. 2B). Upon examining fungal burdens in the brains and lungs of C-IRIS mice, significantly higher fungal load was only detected in the brain, not the lung (Fig. 2C and Supplementary Fig. 2B). Flow cytometric analyses of the brains and lungs revealed increasing numbers of CD4[+] T cells, neutrophils, and macrophages in the brains of C-IRIS mice than those of the three control cohorts; higher numbers of CD4[+] T cells in the lungs of C-IRIS mice than those of the CD4[+] T alone group (Fig. 2D and Supplementary Fig. 3). Taken together the observations of brain pathology and elevated number of CD4[+] T cells in C-IRIS mice, we hypothesize that infiltrating CD4[+] T cells into the brains of CnH99-preinfected mice trigger brain structure abnormalities directly or indirectly via interactions with CnH99 and other immune cells.

To evaluate the role of CD4[+] T cells in the brain, we performed passive CD4[+] T cell transfer through intracerebroventricular (i.c.v.) injection, instead of the peripheral i.v. injection in the regular C-IRIS model, to confirm whether CD4[+] T cells in the brain could trigger C-IRIS. Briefly, CD4[+] T cells were isolated from the lungs of $Tcra^{-/-}$ mice with CD4[+] T cell transfer alone or of C-IRIS $Tcra^{-/-}$ mice (CnH99 preinfection 3 weeks prior and CD4[+] T cell transfer) at seven days post transfer. Then isolated CD4[+] T cells ($4 \times 10^4$ cells) were i.c.v. transferred to the brains of the recipient $Tcra^{-/-}$ mice that had received vehicle

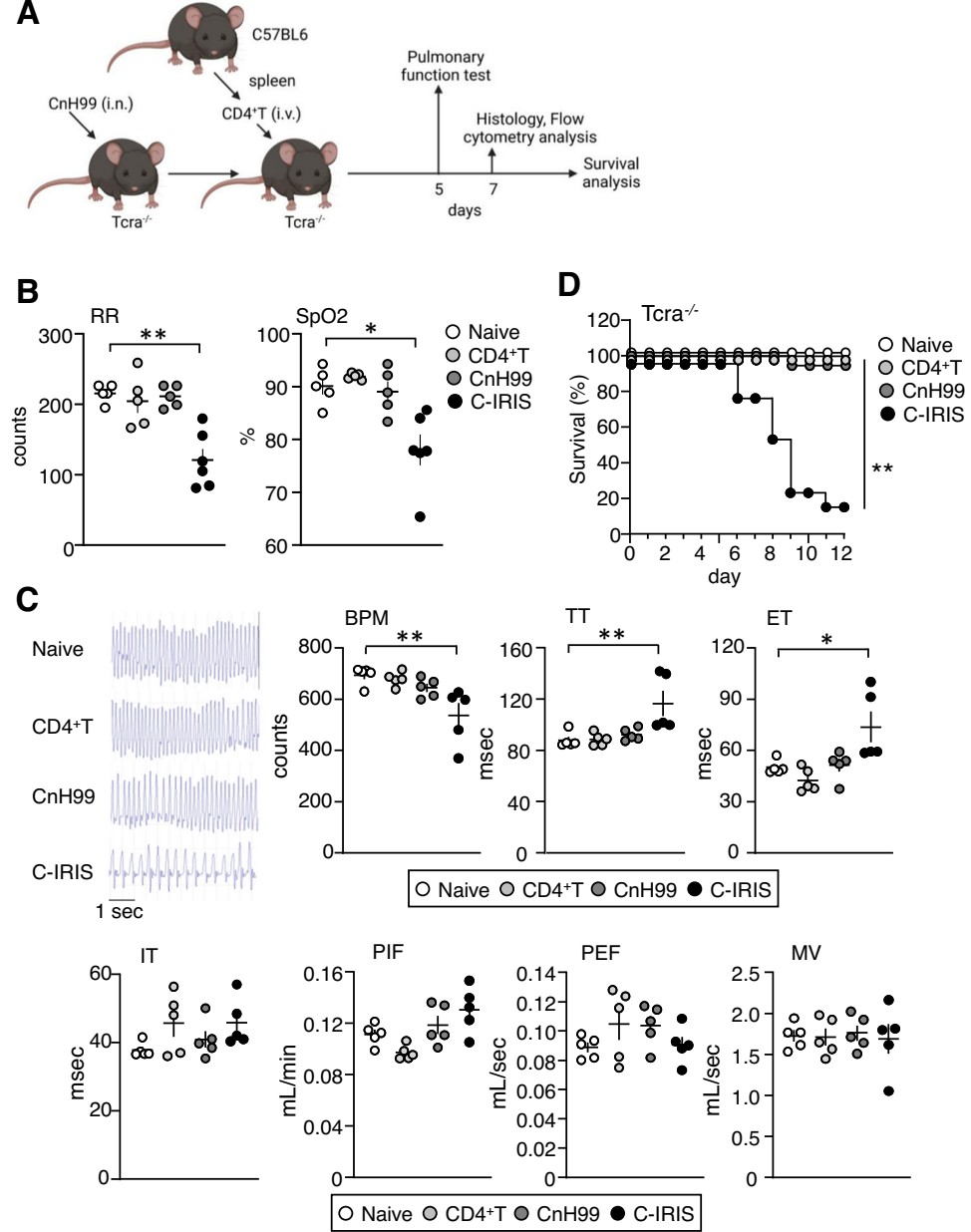

**Fig. 1 | Pulmonary dysfunction and mortality under the C-IRIS condition.**
**A** Schematic representation of the experiments and timeline. Cartoon was created with BioRender.com. **B** Respiratory rates (RR) and oxygen saturation (SpO₂) in four cohorts: *Tcra*⁻/⁻ mice that 1) received neither CnH99 infection nor CD4⁺ T cell transfer, 2) received CD4⁺ T cells, 3) received CnH99 infection, and 4) received CnH99 infection three weeks prior and then CD4⁺ T cells. *n* = 4–6 mice per group. Tukey's multiple comparison test was used for statistical analysis following one-way ANOVA. *$p \le 0.05$. **$p \le 0.01$. **C** Representative plethysmograph and WBP-derived

respiratory measurements: breaths per minute (BPM), total time (TT), expiration time (ET), inspiration time (IT), peak inspiratory flow (PIF), peak expiratory flow (PEF), and minute volume (MV) in the four cohorts. *n* = 5 mice per group. Tukey's multiple comparison test was used for statistical analysis following one-way ANOVA. *$p \le 0.05$. **$p \le 0.01$. **D** Mouse survival from the four cohorts. *n* = 13–14 mice per group for all experiments. Log-rank (Mantel−Cox) test was used for statistical analysis. **$p \le 0.01$. All data are presented as mean values ± SEM.

(i.n.) or CnH99 (100 yeasts, i.n.) 4 weeks prior (Fig. 3A). At day 3 post transfer, we examined pulmonary functions. Among the four groups, only CnH99-preinfected *Tcra*⁻/⁻ recipient mice that i.c.v. received C-IRIS-derived CD4⁺ T cells showed a significant reduction in the respiratory rates and oxygen saturation levels (Fig. 3B). By comparison between CnH99-preinfected recipient mice that had i.c.v. received either C-IRIS CD4⁺ T cells or naive CD4⁺ T cells (isolated from CD4⁺ T cell-transferred *Tcra*⁻/⁻ mice), we also found decreased BPM; elevated TT, ET, and IT; and higher mortality rates in the C-IRIS CD4⁺ T cells recipient group (Fig. 3C, D). These results strongly suggest that CD4⁺ T cells in the brain play a role in pulmonary dysfunction and mortality.

**C-IRIS-derived CD4⁺ T cells mediate neuronal damage in the NTS**
Because NTS in the brainstem is well known to control respiratory functions via sympathetic and parasympathetic systems[22,23], we hypothesized that infiltrated CD4⁺ T cells would trigger neuronal damage in the NTS region and eventually cause pulmonary dysfunction under the C-IRIS disease condition. To confirm this hypothesis, first, we evaluated whether CD4⁺ T cells were distributed to the NTS region. We induced C-IRIS disease in *Tcra*⁻/⁻ mice using CnH99 preinfection and transfer of CD4⁺ T cells, which were isolated from CD4cre-EYFP floxed mice that visualize CD4⁺ T cells as EYFP signal. The presence of EYFP signals in the NTS of C-IRIS mice was confirmed (Fig. 4A). Second, we evaluated the

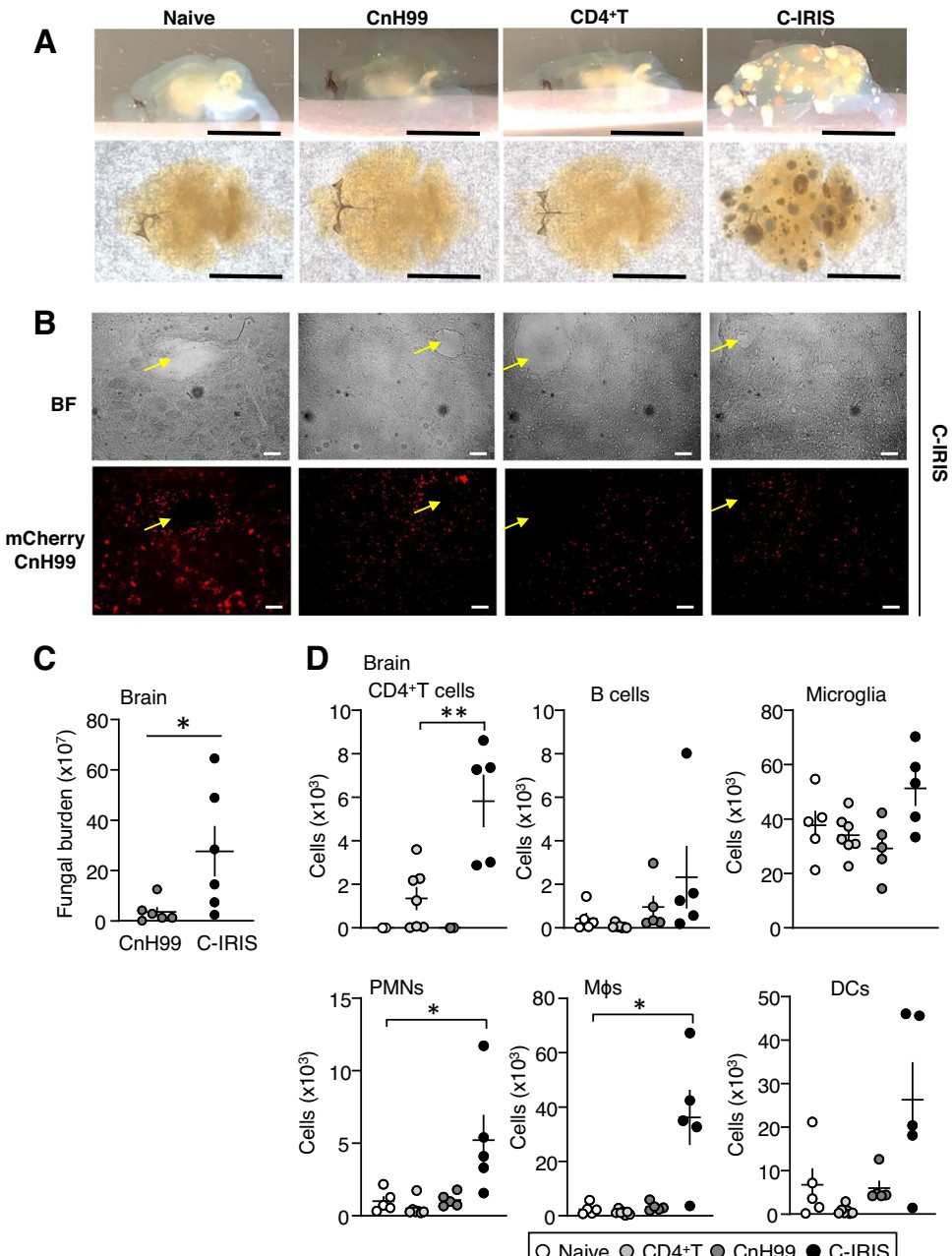

**Fig. 2 | Brain status in C-IRIS mice. A** Representative cleared brain samples from cohorts as indicated in Fig. 1. Scale bar indicates 500 μm. **B** Brightfield (BF) images of the brain and mCherry-CnH99 localization in C-IRIS mice. Yellow arrows indicate tissue damage regions. Scale bar indicates 50 μm. **C** Fungal burden in the brain in CnH99 infection alone and C-IRIS mice. $n = 6$ mice per group. The two-tailed Student's $t$ test was used for statistical analysis. $*p \leq 0.05$. **D** Flow cytometry analyses of brains from cohorts as indicated in Fig. 1. $n = 5-7$ mice per group. Tukey's multiple comparison test was used for statistical analyses following one-way ANOVA. $*p \leq 0.05$. $**p \leq 0.01$. All data are presented as mean values ± SEM.

Fluoro-Jade C staining, which identifies degenerating neurons[24,25], and found that significantly higher amounts of Fluoro-Jade C signals were observed in the NTS region of C-IRIS mice (Supplementary Fig. 4A). Third, we evaluated neuronal numbers and neurite lengths in NTS region using Golgi–Cox silver staining and neuronal morphology in the NTS region using our confocal reflection super-resolution (CRSR) method[26]. Neuron number in the NTS region was significantly decreased in C-IRIS mice (Supplementary Fig. 4B). Neurites were significantly shortened in the NTS region compared with the other three control groups, as indicated above (Fig. 4B, C). Then, to further confirm that CD4[+] T cells derived from C-IRIS mice could directly trigger neuronal damage, we evaluated N2a cellular status after co-culture with CD4[+] T cells. When N2a cells were pre-cultured for three days to extend

neurite and then co-cultured for one day with vehicle, naive CD4[+] T cells, or CD4[+] T cells derived from C-IRIS mice, significant neurite retraction was observed only in the co-culture with CD4[+] T cells derived from C-IRIS mice (Fig. 4D). These results suggest that the capacity of C-IRIS-derived CD4[+] T cells in influencing neurite growth in the NTS and that infiltrated CD4[+] T cells into the brain may trigger neuronal damage under the C-IRIS condition via CD4[+] T cell interactions with neurons.

**Alterations of neuronal network properties in the NTS under the C-IRIS condition**

To understand the implication of NTS neurite retraction, we examined if these neurons had any functional dysregulations. We employed the chemo-genetic manipulation approach of the neuron-activating

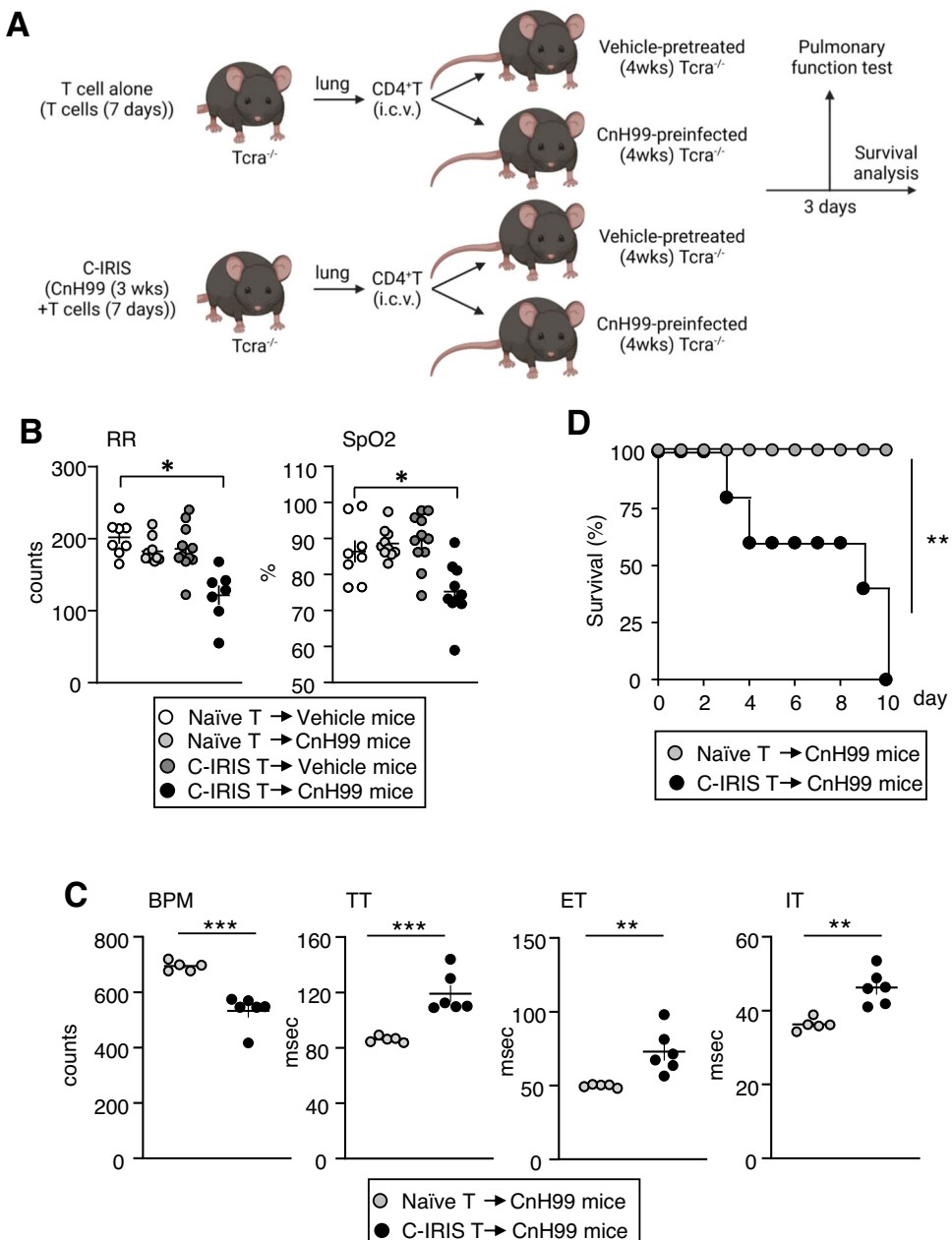

**Fig. 3 | CD4⁺ T cells in the brain trigger pulmonary dysfunction under the C-IRIS condition. A** Schematic model of C-IRIS induction with i.c.v. T cell passive transfer. CD4⁺ T cells were isolated from *Tcra⁻/⁻* mice that received CD4⁺ T cells 7 days prior to isolation (termed naive CD4⁺ T cells) or C-IRIS *Tcra⁻/⁻* mice (termed C-IRIS CD4⁺ T cells). Then, isolated CD4⁺ T cells (4 × 10⁴ cells) were i.c.v. transferred to *Tcra⁻/⁻* mice received vehicle or CnH99 (100 yeasts) four weeks prior to i.c.v injection. Cartoon was created with BioRender.com. **B** Respiratory rates (RR) and oxygen saturation (SpO₂) in four cohorts: (1) vehicle-treated and naive-CD4⁺-T-transferred *Tcra⁻/⁻* mice, (2) CnH99-infected and naive-CD4⁺-T-transferred *Tcra⁻/⁻* mice, (3) vehicle-treated and C-IRIS-CD4⁺-T-transferred *Tcra⁻/⁻* mice, and (4) CnH99-infected and C-IRIS-CD4⁺-T-transferred *Tcra⁻/⁻* mice. *n* = 7–13 mice per group. Tukey's multiple comparison test was used for statistical analysis following one-way ANOVA. *$*p \le 0.05$. **C** WBP-derived respiratory measurements: breaths per minute (BPM), total time (TT), expiration time (ET), and inspiration time (IT) in (1) CnH99-infected and naive-CD4⁺-T-transferred *Tcra⁻/⁻* mice and (2) CnH99-infected and C-IRIS-CD4⁺-T-transferred *Tcra⁻/⁻* mice. *n* = 6–7 mice per group. The two-tailed Student's *t* test was used for statistical analysis. *$*p \le 0.05$. **$**p \le 0.01$. ***$***p \le 0.001$. **D** Mouse survival from cohorts indicated in (**C**). *n* = 5–6 mice per group. Log-rank (Mantel–Cox) test was used for statistical analysis. **$**p \le 0.01$. All data are presented as mean values ± SEM.

Designer Receptors Exclusively Activated by Designer Drugs (DREADDs)[27] and evaluated neuronal activities in NTS downstream neurons by staining of c-Fos, an immediate-early gene and localizes nuclei of neurons upon activation[28]. Figure 5A shows no difference in mCherry expression (by DREADDGq-mCherry AAV) in the NTS between Vglut1-cre-*Tcra⁻/⁻* recipient mice with and without C-IRIS induction (CnH99 and CD4⁺ T cells treatments). c-Fos staining was performed to compare the neuronal activation status of NTS-

downstream medulla respiratory neurons, such as neurons in the rostroventrolateral reticular nucleus (RVL), nucleus ambiguous (Amb), and rostral ventral respiratory group (RVRG), regions known to control pulmonary function[29–32]. c-Fos expressions in the RVL, Amb, and RVRG were significantly reduced in C-IRIS mice compared with the control group (Fig. 5B). These results suggest that neuronal disconnection from NTS regions to downstream respiratory neurons, which may lead to pulmonary dysfunction in mice with C-IRIS.

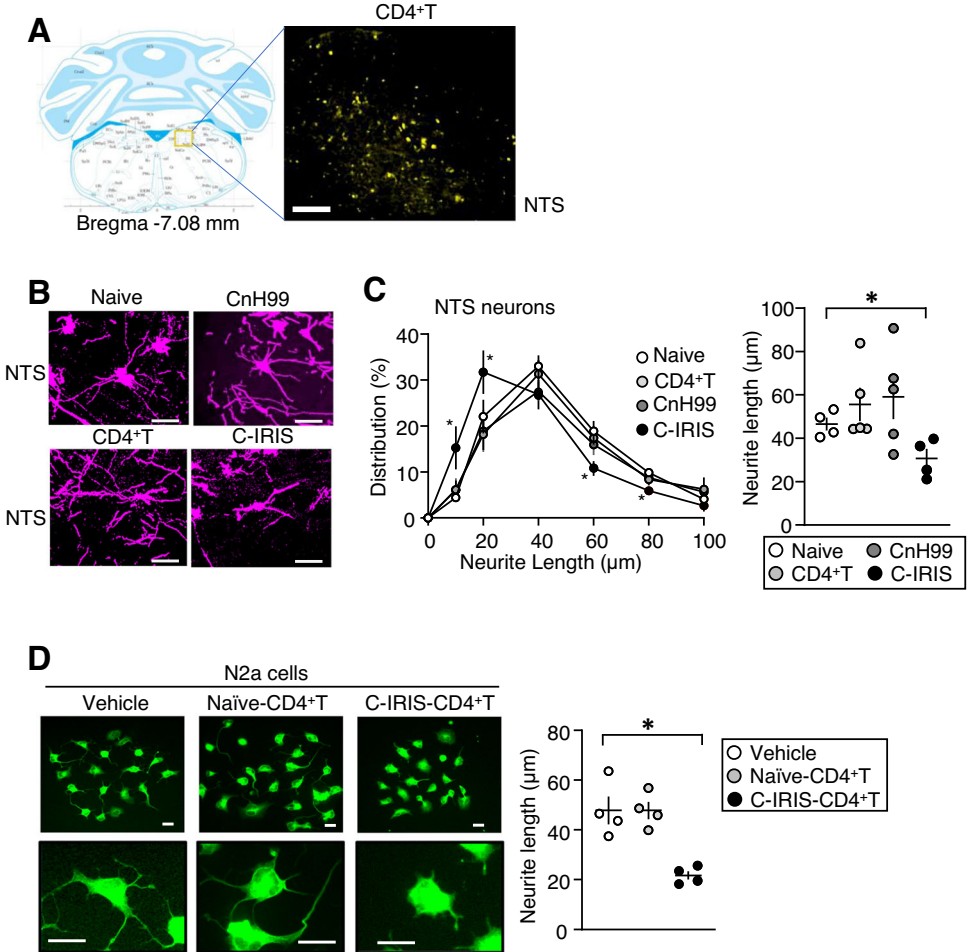

**Fig. 4 | T cell-mediated neuronal damage in the NTS of C-IRIS mice. A** CD4⁺ T cell expression in the NTS region seven days after CD4⁺ T cell transfer to *Tcra⁻/⁻* mice with CnH99 preinfection. Cartoon was created with BioRender.com. Scale bar indicates 200 μm. **B** Representative neuron shapes in the NTS region of four conditions (naive, CnH99 infection alone (CnH99), CD4⁺ T cell transfer alone (CD4⁺ T), and CnH99 preinfection and CD4⁺ T cell transfer (C-IRIS)) using the CRSR neuron analysis method. Scale bar indicates 50 μm. **C** Neurite length population and an average of neurites in the above four groups. *n* = 4–5 mice (at least 20 neurons per each sample were evaluated) per group. Tukey's multiple comparison test was used for statistical analysis following one-way ANOVA. *$p \leq 0.05$. **D** Co-culture between N2a cells and CD4⁺ T cells. CD4⁺ T cells were isolated from *Tcra⁻/⁻* mice that received CD4⁺ T cells 7 days prior isolation (termed naive CD4⁺ T cells) or C-IRIS *Tcra⁻/⁻* mice (termed C-IRIS CD4⁺ T cells). Isolated CD4⁺ T cells were cultured with N2a cells for one day. *n* = 4 biological samples (at least 20 cells per each sample were evaluated) per group. Scale bar indicates 20 μm. Tukey's multiple comparison test was used for statistical analyses following one-way ANOVA. *$p \leq 0.05$. All data are presented as mean values ± SEM. *$p \leq 0.05$.

## CD4⁺ T cells induce neuronal damage via upregulated ephrin B3 and semaphorin 6B

Given that C-IRIS mice showed shortened neurites in the NTS region and C-IRIS-derived CD4⁺ T cells induced neurite retraction (Fig. 4B–D), we evaluated expression levels of ephrin (A1, A2, A5, B1, B2, B3) and semaphorin (3D, 3 F, 4 A, 4D, and 6B), which potently trigger neurite retraction[33–36]. We found that ephrin B3 and semaphorin 6B gene expressions were significantly increased in CD4⁺ T cells isolated from lungs of C-IRIS mice compared to the control group (CD4⁺ T cells transfer alone) (Fig. 6A). Then, we treated ephrin B3 or semaphorin 6B shRNA lentivirus in C-IRIS-derived CD4⁺ T cells to reduce their expression for a co-culture study and an adoptive transfer study (Fig. 6B). Successful reduction was confirmed by flow cytometry and qPCR (Supplementary Fig. 5A, B). Following shRNA treatment, CD4⁺ T cells were co-cultured with N2a cells or i.c.v. transferred to *Tcra⁻/⁻* mice that received CnH99 (100 yeasts, i.n.) 4 weeks prior (same as Fig. 3) to evaluate neuronal conditions in vitro and in NTS; pulmonary function was also evaluated (Fig. 6B). Results obtained from CD4⁺ T cells with ephrin B3 or semaphorin 6B shRNA were compared with those obtained from naive condition and CD4⁺ T cells with control shRNA. In vitro studies revealed that N2a cells co-cultured with ephrin

B3 or semaphorin 6B shRNA-treated C-IRIS-derived CD4⁺ T cells have less neurite retraction, compared with control shRNA treatment (Fig. 6C). Further, when C-IRIS was induced with ephrin B3 or semaphorin 6B shRNA-treated C-IRIS-derived CD4⁺ T cells, significantly higher neurite lengths were observed in the NTS region, compared with control shRNA treatment (Fig. 6D). Furthermore, respiratory rates and oxygen saturation levels were higher as well in C-IRIS mice induced with ephrin B3 or semaphorin 6B shRNA-treated CD4⁺ T cells, compared with control shRNA treatment (Fig. 6E). All these results suggest that potential involvement of ephrin B3 and semaphorin 6B, such that upregulated ephrin B3 and semaphorin 6B in CD4⁺ T cells under C-IRIS may mediate neuronal damage in the NTS, leading to pulmonary dysfunction.

## CCL8-CCR5 axis is crucial for CD4⁺ T cell infiltration to the brain in the C-IRIS mouse model

Because CnH99 is known to enter the brain via direct transmigration of the capillary endothelium or infected phagocytic cells[37,38] and was detected in the brains of *Rag1⁻/⁻* mice at four weeks post CnH99 injection[13], we speculated that infiltrated CnH99 into the brain would upregulate chemokines to attract CD4⁺ T cells from the periphery. To

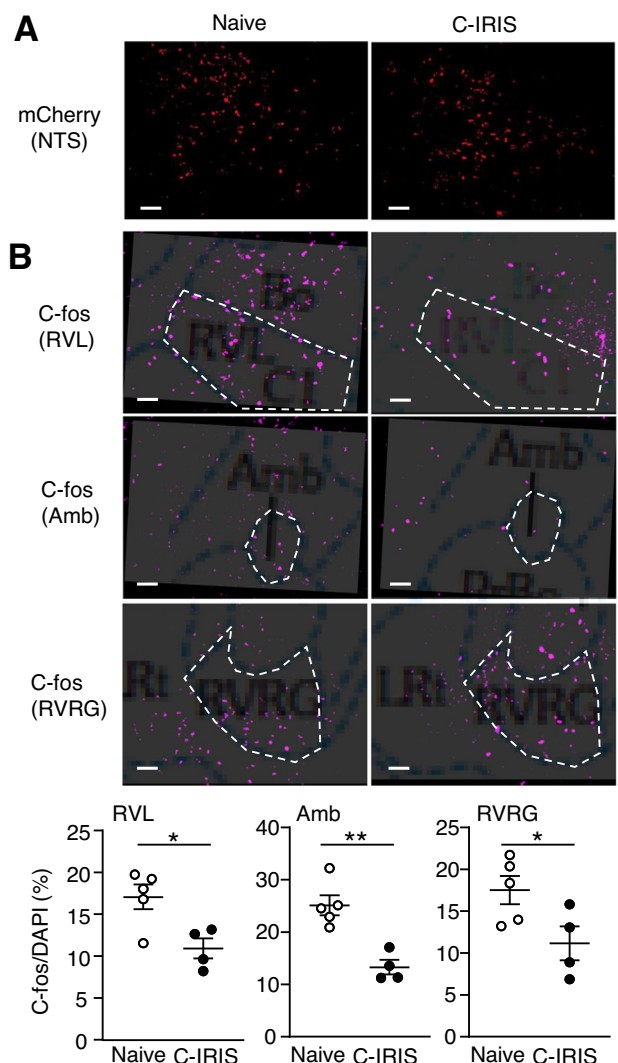

**Fig. 5 | Neuronal network disconnection. A** Representative images of mCherry expression in the NTS region. **B** Representative images and quantification assay of c-fos expression in the RVL, Amb, and RVRG. Brain atlas images were merged with images to identify the region. This process was performed with brain slices including these regions. Scale bar indicates 50 μm. *n* = 4−5 mice per group for all experiments. The two-tailed Student's *t* test was used for statistical analysis. *$p \leq 0.05$. **$p \leq 0.01$. Data are presented as mean values ± SEM. *$p \leq 0.05$.

confirm this possibility, we evaluated chemokine expression levels in the hindbrain, including the NTS region, of vehicle-treated or CnH99-infected mice (CnH99 alone) using a chemokine protein array. Four weeks after CnH99 infection or vehicle treatment into *Tcra*[−/−] mice, Chemokine C-C motif ligand 8 (CCL8), an endogenous ligand for C-C Motif Chemokine Receptor 5 (CCR5), was significantly upregulated (Fig. 7A). Moreover, CCR5 was upregulated on CD4[+] T cells isolated from lungs of C-IRIS in *Tcra*[−/−] mice (CnH99 3 weeks prior + CD4[+] T cells 1 week), compared with CD4[+] T cell transfer alone (for 1 week) in *Tcra*[−/−] mice (Fig. 7B). To examine the role of CCR5 in C-IRIS induction, we induced C-IRIS with either wild-type CD4[+] T cells derived from C57BL6 mice or CCR5[−/−] CD4[+] T cells derived from C57BL6-background CCR5[−/−] mice. A reduced number of CD4[+] T cells was noted in the brains of mice when C-IRIS was induced with CCR5[−/−] CD4[+] T cells (Fig. 7C); in the same group, increased neurite lengths in the NTS were also noted (Fig. 7D). WBP from *Tcra*[−/−] mice with CCR5[−/−]-CD4[+]-T-induced C-IRIS indicated significantly higher BPM, lower TT and ET, compared with those in *Tcra*[−/−] mice with wild-type CD4[+]-T-induced C-IRIS (Fig. 7E). To

evaluate the role of the CCL8-CCR5 axis in the C-IRIS disease, daily intraperitoneal (i.p.) administration of Maraviroc, a CCR5-specific antagonist, at a dose of 5 mg/kg was performed alongside the injection of CD4[+] T cells into the brain of CnH99 preinfected *Tcra*[−/−] mice. We confirmed that infiltration of CD4[+] T cells into the brain was significantly reduced in the Maraviroc-treated group compared with the vehicle-treated control group (Fig. 7F). Respiratory rates and oxygen saturation levels were improved by Maraviroc treatment (Fig. 7G) and increased neurite lengths in the NTS were detected (Fig. 7H). Expression of ephrin B3 and semaphorin 6B was evaluated and found unchanged on CD4[+] T cells between the Maraviroc-treated group and the vehicle-treated control group (Supplementary Fig. 6). These results suggest that the CCR5 signal may contribute to CD4[+] T cell infiltration, but not CD4[+] T cell neurotoxic property change, which triggers the C-IRIS phenomenon, and Maraviroc may be a potential therapeutic option in C-IRIS disease.

## Discussion

Despite increasing clinical reports on C-IRIS, mechanistic understanding of IRIS is still largely limited. One reason is the lack of a representative experimental model. Recently, we established a mouse model of unmasking C-IRIS based on lymphocyte-deficient *Rag1*[−/−] mice immune reconstituted by adoptive transfer (i.v.) of CD4[+] T cells after CnH99 infection[13], and mice with this C-IRIS induction method presented with high mortality rates, infiltration of T cells to the brain, systemic upregulation of pro-inflammatory cytokines, and brain edema[13]. In the present study, we built upon our previous findings and sought to identify the underlying mechanisms of pulmonary dysfunctions. C-IRIS was induced in the immunocompromised *Tcra*[−/−] mice (lack T cells only), considering low baseline CD4[+] T cells in the clinical condition. We demonstrated that *Tcra*[−/−] mice that received CnH99 (100 yeasts, i.n.) preinfection and CD4[+] T cell (10[6] cells, i.v.) transfer (C-IRIS condition) also had a significantly higher mortality rate than that of control mice (naive, CD4[+] T cell transfer alone, CnH99 infection alone). In addition, we demonstrated that the Cn52D was inefficient for induction in the same treatment condition, as seen from the non-significant results of survival analysis and respiratory rate measurement, which concurs with the previous report that used another Cn serotype D strain, Cn1841[39]. We suggest using CnH99 in experimental C-IRIS induction to recapitulate the C-IRIS condition. However, it would be an interesting future topic to examine the efficacy of Cn52D with a high dose and different induction paradigm.

C-IRIS patients can present with pulmonary disease, for instance, respiratory distress, pulmonary nodules and pleural effusions[3,8,12]. In this study, we analyzed pulmonary functions via SomnoSuite and WBP. Oxygen saturation, or SpO₂, and respiratory rates were significantly lower in C-IRIS group only, indicating potential hypoxia and respiratory rate depression. A decrease in respiratory rates could also stem from ankylosis; however, because oxygen saturation was also decreased in C-IRIS mice, suggesting ankylosis may be not involved in the reduced respiratory rates in C-IRIS mice. Additionally, given that respiratory rates from SomnoSuite were obtained in anesthetized conditions, we also performed unrestrained and unanesthetized WBP. BPM remained reduced in C-IRIS mice. The shape of the respiratory cycle is helpful for understanding breathing patterns, and disturbances in cycle shape have been utilized to explore how neuronal functions may control respiratory drives and reflexes[40,41]. We found C-IRIS mice also showed abnormal respiration; TT and ET showed an increase in time, while no significant changes were found in PIF, PEF, or MV. Both TT and ET are measures of the duration of a respiration cycle, and a prolonged time for expiration and completion of one ventilatory process may suggest labored breathing under the C-IRIS condition.

Given minimal injury found in the lung tissues but cerebral abnormality in the brain was detected[13], we speculated that the brain is more favored by pathological changes. In the brains of C-IRIS mice,

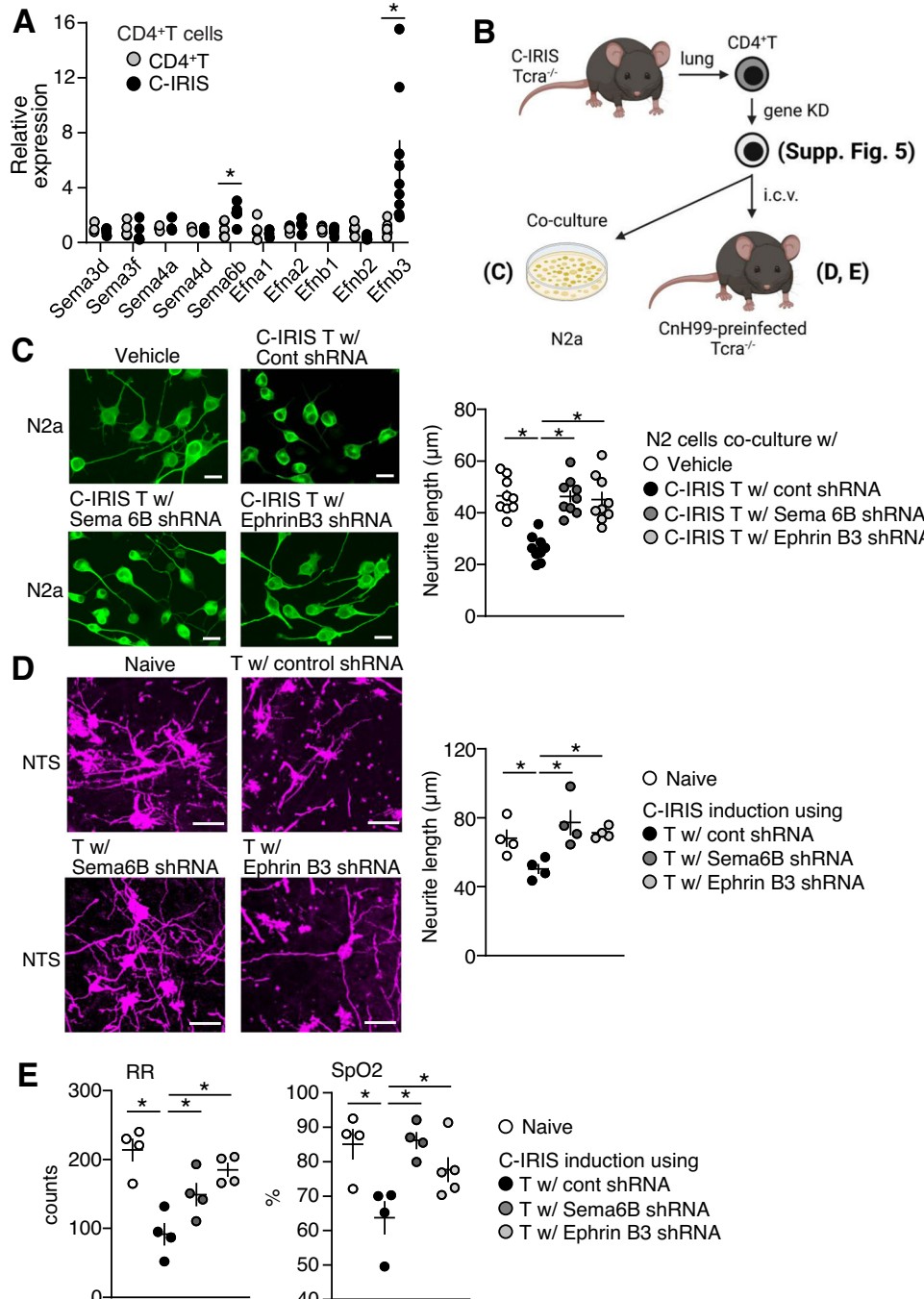

**Fig. 6 | Critical molecules in neuronal damage in the NTS regions of C-IRIS mice.** **A** qPCR analysis. CD4+ T cells were isolated from *Tcra*−/− mice that received CnH99 infection for three weeks and then CD4+ T cells for seven days (C-IRIS) or that received only CD4+ T cell transfer for seven days. $n = 4–10$ mice per group. The two-tailed Student's *t* test was used for statistical analysis. *$p ≤ 0.05$. **B** Schematic model for the studies in (**C**–**E**) and Supplementary Fig. 5. CD4+ T cells were isolated from C-IRIS *Tcra*−/− mice (same condition in (**A**)). According to the manufacturer's protocol, isolated CD4+ T cells were treated with control, Semaphorin 6B, or Ephrin B3 shRNA for three days. The shRNA-treated CD4+ T cells were used for experiments (**C**–**E**). Cartoon was created with BioRender.com. **C** N2a cell-CD4+ T cell coculture. shRNA-treated T cells were cultured with N2a cells for 24 h. Neurite length population and average were evaluated. $n = 9$ biological samples per group. Around

20 cells were evaluated in each sample. Scale bar indicates 20 μm. The two-tailed Student's *t* test was used for statistical analysis (compared with control shRNA group). *$p ≤ 0.05$. **D** The shRNA-treated CD4+ T cells were i.c.v. transferred to the CnH99-treated *Tcra*−/− mice. At seven days after transfer, neurite length population and average were evaluated under the CRSR analysis. $n = 4$ mice per group. Around 20 neurons were evaluated in each sample. Scale bar indicates 50 μm. The two-tailed Student's *t* test was used for statistical analysis (compared with control shRNA group). *$p ≤ 0.05$. **E** Three days after transfer, respiratory rates and oxygen saturation were evaluated. $n = 4–5$ mice per group. The two-tailed Student's t test was used for statistical analysis (compared with control shRNA group). *$p ≤ 0.05$. All data are presented as mean values ± SEM.

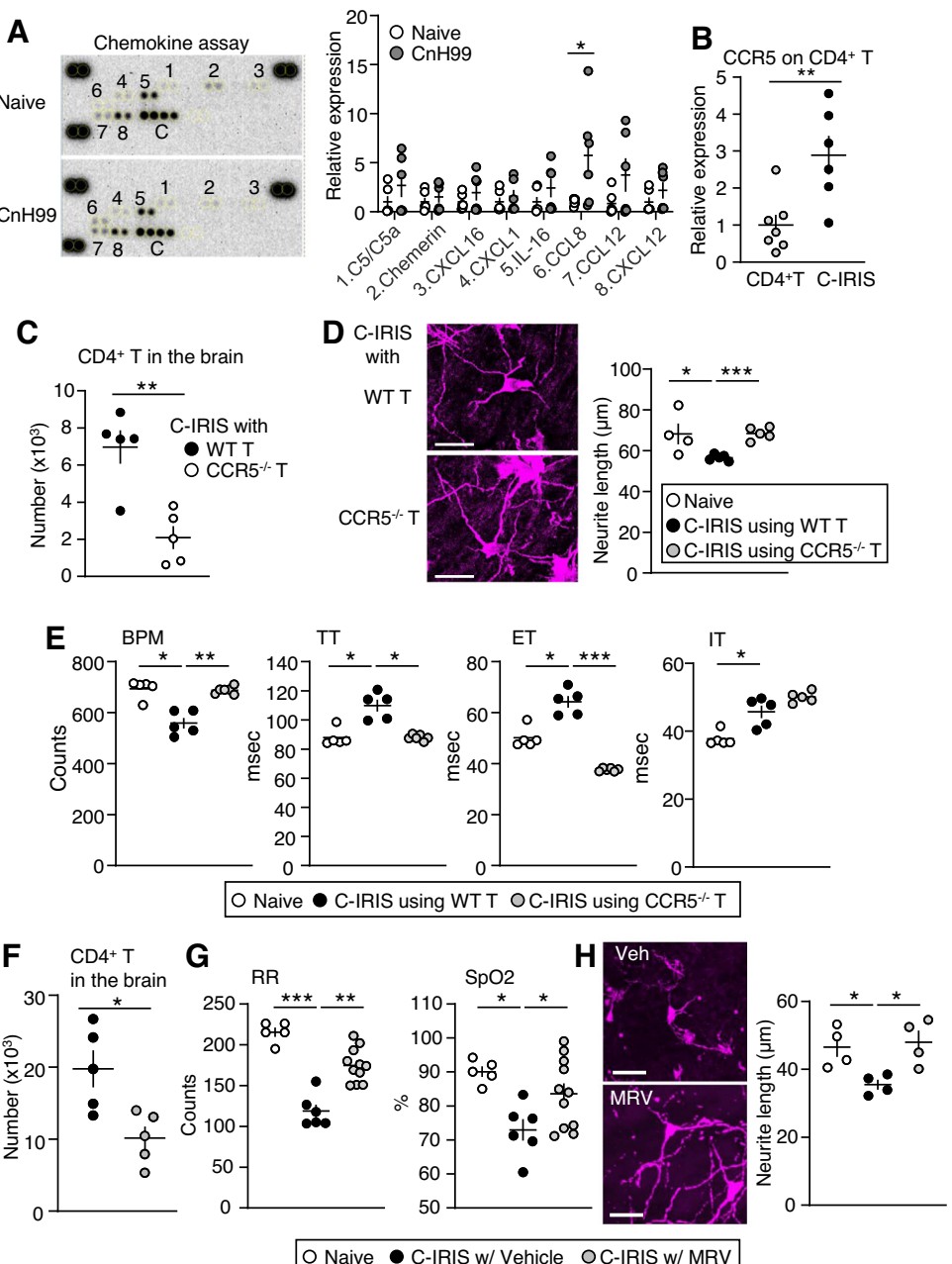

**Fig. 7 | Critical chemokine signal for T cell migration and eventual neuronal damage. A** Chemokine array. Hindbrains were isolated from *Tcra*⁻/⁻ mice that received vehicle or CnH99 four weeks prior. *n* = 5 mice per group. The two-tailed Student's *t* test was used for statistical analyses. *\*p ≤ 0.05*. **B** CCR5 mRNA expression in CD4⁺ T cells from lungs of *Tcra*⁻/⁻ mice that received CD4⁺ T cells for seven days with (C-IRIS) and without (CD4⁺ T alone) CnH99 preinfection three weeks prior. *n* = 6–7 mice per group. The two-tailed Student's *t* test was used for statistical analysis. *\*\*p < 0.01*. **C** CD4⁺ T cell number in the brain in C-IRIS mice induced with wild-type CD4⁺ T or CCR5⁻/⁻ CD4⁺ T cells. *n* = 5 mice per group. The two-tailed Student's *t* test was used for statistical analysis. *\*\*p < 0.01*. **D** Neurite lengths in the NTS in naive mice and C-IRIS mice induced with wild-type CD4⁺ T or CCR5⁻/⁻ CD4⁺ T cells. We used naive data as a reference. Naive data is the same as Fig. 6D. Scale bar indicates 50 μm. *n* = 5 mice per group. The two-tailed Student's *t* test was used for statistical analysis (compared with WT T cells group). *\*p ≤ 0.05*. *\*\*\*p ≤ 0.001*. **E** WBP-derived respiratory measurements: breaths per minute (BPM), total time (TT), expiration time (ET), and inspiration time (IT) in naive mice and C-IRIS mice induced with wild-type CD4⁺ T or CCR5⁻/⁻ CD4⁺ T cells. Naive data is the same as Fig. 1C. *n* = 5 mice per group. The two-tailed Student's *t* test was used for statistical analysis (compared with WT T group). *\*p ≤ 0.05. \*\*p ≤ 0.01. \*\*\*p ≤ 0.001*. **F** CD4⁺ T cell numbers in the brain in C-IRIS mice treated with vehicle or Maraviroc (MRV). *n* = 5 mice per group. The two-tailed Student's *t* test was used for statistical analysis. *\*p < 0.05*. Inhibitor functions of MRV on respiratory rates and oxygen saturation (**G**) and neuronal damage in the NTS regions (**H**). *n* = 5–11 mice per group for (**G**). *n* = 4 mice per group for (**H**). Naive data is the same as Figs. 1B and 4C, respectively. MRV or vehicle (PBS) was treated every day from CD4⁺ T cell transfer into *Tcra*⁻/⁻ mice that received CnH99 preinfection for three weeks. Brains for flow cytometry and the CRSR analysis and respiratory rate measurements were conducted seven days after CD4⁺ T cell transfer. Scale bar indicates 50 μm. The two-tailed Student's *t* test was used for statistical analysis (compared with vehicle group). *\*p ≤ 0.05*. All data are presented as mean values ± SEM.

abnormal tissue morphology and colonization of Cn in multiple regions were evident, consistent with previous pathology reports[20]. These findings were accompanied by the discovery of escalated levels of fungal load, CD4[+] T cells, neutrophils, and macrophages in the brains. Because these abnormal tissue morphology and colonization of Cn were only observed under the C-IRIS condition, CD4[+] T cells in the brain may influence Cn distribution and tissue damage. The increase in fungal burden in the brain, not the lung, may further suggest a particular susceptibility of the brain to IRIS-related neurological changes, and fungal proliferation may challenge host integrity, contributing to the disease phenotype. Because our model is a CD4[+] T cell-driven unmasking C-IRIS disease, we investigated CD4[+] T cell function in generating an inflammatory state in the present study. However, the role of other immune cells, such as monocytes, neutrophils, and microglia, is not well studied in the field, motivating future studies looking at their involvement and possible contributions to CD4[+] T cell property changes (e.g., CCR5 and neurotoxic molecules) and CD4[+] T cell-mediated neurodegeneration.

Previously we reported hindbrain edema in C-IRIS mice[13]. The brainstem contains nuclei that control vital life functions, such as breathing, consciousness, blood pressure, heart rate, and sleep[42]. Here we found that neurite lengths in the NTS of the C-IRIS group were significantly decreased compared to those of the control groups. The shortened neurites in the NTS have led us to propose a disruption in the neuronal network initiated by this region. Neurons in the NTS are implicated in inspiratory respiration via activation of the Amb, RVRG, and RVL[29–32]. Specifically, activating NTS glutamatergic neurons are involved in sustaining baseline signaling for respiration[43–45]. Optogenetic and chemogenetic approaches help investigate neuronal networks[46,47]. In the present study, we employed the chemogenetic method to show that NTS neurons disconnected to downstream neurons in the Amb, RVRG, and RVL under the C-IRIS condition, which may be due to shortened neurites of NTS neurons. We demonstrated the neuronal damage property of C-IRIS-derived CD4[+] T cells using the co-culture system with neurons and i.c.v. transfer in vivo system. In sum, damage to respiration-controlled neuronal functions in the NTS may be a potential mechanism for pulmonary dysfunction under the C-IRIS condition in mice.

The NTS also projects to the hypothalamus, interpeduncular nucleus (IPN), and parabrachial nucleus (PBN) in the brain[48–50]. The NTS-hypothalamus axis is known to control food intake behaviors[51]. Because NTS neurons have been shown to have neurite reductions, it is also possible that a network disconnection in the NTS-hypothalamus axis exists. Severe losses of body weight (over 20%) were found in *Rag1*[−/− 13] and *Tcra*[−/−] mice with C-IRIS. Therefore, the lack of food intake due to the NTS-hypothalamus network disconnection may be involved in body weight reduction and mortality. Several clinical cases with Cn infections and C-IRIS have reported the presence of poor appetite and/or loss of body weight[52–54]. In addition, the NTS-IPN and NTS-PBN axes are known to be involved in reward and pain sensitivity, respectively[48–50]. Thus, it is also possible that C-IRIS mice show impaired or loss of activities in these functions. These hypotheses require further investigation.

In the present study, we also identified the potential molecular mechanisms by which CD4[+] T cells infiltrate into the brain and induce neuronal damage. Regarding the former mechanism, we found the critical role of the CCL8-CCR5 axis. CnH99 is known to enter the brain via direct transmigration through the capillary endothelium[37] or infected phagocytic cells[38]. Four weeks after CnH99 infection, fungal burden was observed in the brains of the infected mice[13]. Among 25 tested major chemokines in the hindbrain region of CnH99 infected mice four weeks after infection, only the CCL8 level was significantly increased compared to naive mice. It has been reported that CCL8 mRNA is elevated in whole blood samples in patients who developed C-IRIS after ART treatment compared to those who did not develop

C-IRIS[55]. It also has been reported that the expression of CCL8 is elevated when heat-killed Cn was stimulated with PBMC for 24 h and examined by RNA-seq[56]. CCL8, a ligand for CCR5 and CCR1, is expressed in astrocytes and is involved in disease induction via effector CD4[+] T cell migration[57]. CCR5 is expressed on CD4[+] T cells, particularly Th1 lymphocytes, and is involved in viral infections and tumor growth[58,59]. We previously reported that Th1 cells are critical for the C-IRIS induction[13], and here we found upregulation of CCR5 in CD4[+] T cells under the C-IRIS condition. Strikingly, using CCR5[−/−] CD4[+] T cells and Maraviroc, a CCR5-specific antagonist, we found the prevention of CD4[+] T cell migration into the brain and restoration of pulmonary function and neurite growth in the NTS. It has been reported that Maraviroc prevents symptoms in patients with C-IRIS[60]. This clinical evidence supports our findings that CD4[+] T cells migration into the brain via the CCL8-CCR5 axis may be important for C-IRIS. Regarding the latter mechanism, we found that CD4[+] T cells derived from C-IRIS mice had significantly upregulated ephrin B3 and semaphorin 6B levels. Together with their known role in inhibiting neurite development[61], we speculated that ephrin B3 and semaphorin 6B might be involved in neurite retraction observed in C-IRIS mice. Indeed, when selectively knocked down in CD4[+] T cells via shRNA transfection, neuronal damage in the NTS was prevented. Furthermore, in the co-culture of neurons and CD4[+] T cells with shRNA transfection of ephrin B3 and semaphorin 6B, recovery of neurite outgrowth was also confirmed. Moreover, respiratory rates were also significantly recovered in mice that received ephrin B3 shRNA- or semaphorin 6B shRNA-treated CD4[+] T cells, suggesting neuronal damage in the NTS may link to respiratory dysfunction.

Although our mouse model mimics human C-IRIS, the inherent differences in biology and immune responses between the mouse C-IRIS model and human C-IRIS situation should not be ignored. For instance, C-IRIS patients develop respiratory failure at the point of brain/uncal herniation[12]. However, although our C-IRIS model mice show brain edema[13], which may trigger brain herniation, such mice do not show brain/uncal herniation when mice develop respiratory failure. The basis for these differences is not understood and merits further investigation. It is unlikely that any single animal model will fully recapitulate the human disease in all its details. Thus, developing other C-IRIS animal models that overcome current limitations or mimic paradoxical C-IRIS is essential. Further investigation of IRIS pathogenesis using our and future models would lead to developing therapies for C-IRIS.

In conclusion, we illustrate that pulmonary dysfunction associated with the C-IRIS condition in mice may be attributed to the infiltration of CD4[+] T cells into the brain via the CCL8-CCR5 axis, which triggers NTS neuronal damage and neuronal disconnection via upregulated ephrin B3 and semaphorin 6B in CD4[+] T cells. Currently, non-steroidal anti-inflammatory drugs (NSAIDs) and corticosteroids are prescribed to suppress excessive inflammation in C-IRIS patients[62]. However, such immunosuppressive medications may impair the immune response to existing infections and render patients more vulnerable to contracting new infections[63,64]. Collectively, we propose that the prevention of CCR5-specific CD4[+] T cell migration and/or regulation of ephrin B3 and semaphorin 6B functions could be promising therapeutic candidates for the treatment or the prevention of C-IRIS.

## Methods
### Ethics statement
C57BL6, *Rag1*[−/−] (002216), *Tcra*[−/−] (002116), CCR5[−/−] (005427), Slc17a7-IRES2-Cre-D knock-in mice (Vglut1-cre, 023527), Cd4-cre (022071), and floxed EYFP (7903) mice purchased from The Jackson Laboratory were used for the current work. Mice were group-housed with two to five in an individual cage and kept under specific pathogen-free conditions with a 12-h light/dark cycle. Sterile pelleted mouse diet and water were

 

given ad libitum for the health monitoring of mice. Healthy male and female mice aged 16–20 weeks were randomly selected for this study. All studies and experiments were performed under the approval of the Institutional Animal Care and Use Committee (protocol numbers 19171 and 22140) at the University of Illinois.

## C-IRIS induction

*Cryptococcus neoformans* (Cn) serotype A strain H99 (CnH99, ATCC 208821) and *Cryptococcus neoformans* (Cn) serotype D strain 52 (Cn52D, ATCC 24067) were used from fresh isolates from frozen stocks without serial plating. CnH99 and Cn52D were cultured on yeast extract-peptone-dextrose (YPD) medium (yeast extract 1%, peptone 2%, dextrose 2%) plates at 30 °C for 48 h. An isolated CnH99 or Cn52D colony was incubated on YPD liquid medium at 30 °C with 200 rpm shaking overnight. CD4+ T cells were isolated from the spleen and inguinal/axillary lymph nodes of naive C57BL6 mice (6–8 weeks old). Through the negative selection, cells were labeled with biotinylated antibodies for CD19 (Cat# 115504, BioLegend), CD11c (Cat# 117304, BioLegend), CD8 (Cat# 100704, BioLegend), Ly6G (Cat# 127604, Bio-Legend), and CD11b (Cat# 101204, BioLegend) and streptavidin-coated magnetic particles (STEMCELL, Vancouver, Canada, EasySep™ Strep-tavidin RapidSpheres™ Isolation Kit, Cat# 19860 A). B, CD8+ T cells, neutrophils, dendritic cells, and macrophages were separated. Then, through the positive selection using biotinylated CD4 antibody (Cat# 100404, BioLegend) and Biotin-coated magnetic particles (STEMCELL, EasySep™ Biotin Positive Selection Kit, Cat# 17665), CD4+ T cells were isolated. *Tcra*⁻/⁻ and *Rag1*⁻/⁻ male mice aged 16–20 weeks old were anesthetized with isoflurane (5% for induction, 3% for maintenance) in oxygen (2 L/min) and intranasally infected with CnH99 or Cn52D (100 yeasts in 30 μl PBS). The C-IRIS model was induced by intravenous (i.v) injection ($10^6$ cells in 200 μl PBS with 2% FBS) of isolated CD4+ T cells into Cn-infected mice 3 and 4 weeks after CnH99. Furthermore, to examine the effects of T cells in the brain, CD4+ T cells were isolated from lung tissues of *Tcra*⁻/⁻ mice that received CnH99 infection three weeks prior and then CD4+ T cells for seven days or CD4+ T cells only. Then, CD4+ T cells isolated from lungs were intracerebroventricularly (i.c.v.) injected ($4 \times 10^4$ cells in 5 μl PBS with 2% FBS) into CnH99 pre-infected mice 4 weeks after CnH99 infection.

## Pulmonary function test

Mice were anesthetized with 2.0% isoflurane in room air at 500 ml/min until they were less likely to respond to external stimuli. The anesthetized animal was placed in a prone position and continuously induced with 0.3% isoflurane in room air at 30 ml/min (Kent Scientific Somno-Suite). A pulse oximeter was placed on a hind paw to obtain respiratory rate and saturated oxygen level (Kent Scientific PhysioSuite).

## Whole body plethysmography

Experimental mice were acclimated in the room and placed in the testing chamber under no anesthesia or restraint for recording for 5 min using the Buxco Small Animal Whole Body Plethysmography apparatus (Data Sciences International). The apparatus consists of a calibration unit, a flow generator, and a testing chamber. The testing chamber was cleaned with 70% ethanol and let dry between trials. Respiratory parameters were recorded by the device and the data were analyzed using Ponemah® Software, Ver 5.2.

## Tissue preparation

Mice were anesthetized with isoflurane and transcardially perfused with Tris-buffered saline (TBS) followed by 4% paraformaldehyde (PFA). The brains were post-fixed in 4% PFA at 4 °C overnight and then placed in 30% sucrose for cryoprotection until the tissues sank. The brains were embedded in Tissue-Tek OCT Compound, frozen, and cryostat serially sectioned at 30 *μ*m thickness. Slices were collected and preserved in 0.02% azide solution at 4 °C.

## Whole-brain clearing and imaging

The whole-brain clearing was performed using modifying FDISCO[65]. The samples were dehydrated with 50% (v/v), 70% (v/v), 80% (v/v), and 100% (v/v) THF (Sigma-Aldrich, St. Louis, MO) solutions, with 24 h each step at 4 °C. Pure DBA (Sigma-Aldrich) was used as a refractive index matching solution to clear tissue after dehydration. Images of the cleared whole brain were obtained by UltraMicroscope (Miltenyi Bio-Tec, Bielefeld, Germany). The ImageJ (NIH) and Imaris (Bitplane, AG) were used for image processing.

## N2a cells and T cells co-culturing

N2a cells ($1 \times 10^3$ cells/well) were cultured on coverslips in 24-well plates in 1% FBS/DMEM to initiate neuron differentiation[66]. After three days, CD4+ T cells isolated from the lungs of *Tcra*⁻/⁻ mice with CD4+ T cell transfer for 7 days or of C-IRIS mice at 7 days after CD4+ T cell transfer were added to N2a cell culture triplicates at a 3:1 cell ratio and incubated for 24 h at 37 °C with 5% carbon dioxide circulation in a sterile incubator.

## Immunofluorescence analysis

Cultured cells were post-fixed on coverslips using 4% PFA and per-meabilized with Triton-X for 15 min at room temperature. Non-specific sites were blocked with 3% BSA in TBS with 0.1%Tween20 (TBST) for 1 h at room temperature and incubated overnight with primary rabbit anti-mouse beta-3 tubulin antibody (Novus Biologicals, Cat# NBP130048; 1:200). Bound antibodies were visualized using Alexa Fluor 488 anti-rabbit IgG antibody (1:500 dilution; Thermo Fisher Scientific) incubation for 1 h at room temperature, and counterstained with 4′,6-diamidino-2-phenylindole (DAPI) staining for 1 min. Cover-glasses were then mounted with Prolong Diamond Antifade Mountant (Prolong Gold Antifade Mountant, Cat# P36930, Invitrogen). Slides were visualized using the Leica DFC 3000G, and the GFP channels were selected for visualization at ×20 magnification. Analysis of neurite lengths was performed using the Measurement function on ImageJ FIJI.

## Fluoro-Jade C staining

Isolated brain (10 μm thickness) from CnH99-infected and C-IRIS mice was stained by Fluoro-Jade C solution (AAT Bioquest, 23062), according to manufacturer's protocol. Fluoro-Jade C stained neuron was counted in the NTS region.

## Golgi staining and CRSR

Isoflurane deeply anesthetized mice were perfused with 4% PFA via transcardial perfusion. According to the manufacturer's instructions, Golgi-Cox staining was done as described by FD Rapid Golgi-Stain Kit (FD Neuro-Technologies, Inc.). Briefly, brains were harvested and transferred into 10 ml of Solution A and B at a ratio of 1:1 for 24 h, followed by another 14 days in the dark in the same amount of Solution A and B. Brains were then placed in solution C for three days, with a new solution being substituted after 24 h. Then, brains were embedded in Tissue-Tek OCT compound, coronally cut into 60 μm sections, stored in 0.02% azide solution at 4 °C, and mounted on poly-L-lysine coated glass slides. At least 20 sections were mounted and imaged from each animal. Images were taken using the Nikon A1 confocal scanning microscope, with settings selected as CRSR (minimized pinhole at 0.9 AU), at x20 magnification, and 405 nm wavelength[26]. Settings for z-stack were set for the step size of 0.575 μm for all pictures acquired. Neurite length analyses were performed using the NIS software, where the neurites were traced by the polyline tool in measurement.

## Fungal burden

Mice were euthanized by $CO_2$ inhalation. Brains and lungs were immediately extracted and homogenized in 1 ml PBS. Homogenates were serially diluted, plated on YPD plates, and incubated at 30 °C for 48 h to enumerate colony-forming units (CFU).

## Adeno-associated virus (AAV) injections in mice and induction of C-IRIS

Vesicular glutamate transporter (Vglut1) cre-$Tcra^{-/-}$ mice were deeply anesthetized with 2.0% isoflurane and then prepped on a stereotaxic apparatus (Stoelting 51730) with continuous induction with 1.5% iso-flurane (Kent Scientific SomnoSuite) and were stereotaxically injected with Cre-dependent DREADDGq AAV (AAV2-hM3D(Gq)-mCherry, Addgene 44361, $6.5 \times 10^{12}$ unit/ml,150 nl[67] into the NTS. The pAAV-hSyn-DIO-hM3D(Gq)-mCherry (AAV2) was provided by Dr. Bryan Roth (The University of North Carolina at Chapel Hill) under the MTA. Based on the corresponding mouse brain atlas, the following coordinates were applied: AP = −6.96, ML = −0.75, and DV = −3.80. 150 nl of AAV was injected at a 30 nl/min flow rate. Because AAV2-hM3D(Gq)-mCherry is a Cre-dependent excitatory hM3D(Gq) fused with mCherry, NTS glutamatergic neurons were marked by mCherry staining three weeks after injection[68]. Three weeks after AAV injection, we infected CnH99 (100 yeasts, i.n.) into such mice and then transferred CD4+ T cells ($10^6$ cells, i.v.) isolated from B6 mice three weeks after CnH99 infection. Seven days after CD4+ T cell transfer, clozapine-N-oxide (CNO, 1 μg/5 μl, i.c.v.)[67,69] was given to recipient mice to enhance Vglut1-mediated glutamatergic signaling in the AAV2-hM3D(Gq)-mCherry transfected NTS region. Brains were collected 60 min after the CNO injection.

## Immunohistochemistry and analysis for brain neuronal network

Neuronal network activity was analyzed by imaging immediate-early gene c-Fos expression[28]. Coronal sections (30 μm) of the medulla (Bregma −6.6 mm ~ −7.6 mm) were mounted onto poly L-lysine-coated glass slides. The samples were permeabilized with 0.5% Triton-X for 15 min at room temperature, blocked with 2% BSA for 1 h, and incubated overnight at 4 °C with rabbit c-Fos primary antibody (1:500, Cell Signaling Cat# 2250) diluted in 2% BSA in TBST, incubated with donkey anti-rabbit Alexa 647 secondary antibody (1:500) for 2 h at room temperature, and incubated with DAPI for 1 min at room temperature. The slides were mounted using Prolong Gold Antifade Mountant (Cat# P36930, Invitrogen), and coverslipped. Tissue sections were visualized using the Leica DFC 3000G. All image acquisitions were processed using the same laser intensity, exposure, and gain values. mCherry signals in the NTS region were visualized using a 20X objective. DAPI and c-Fos signals in ROIs were visualized using a 10X objective. ImageJ software was used to count DAPI and c-Fos signals. Before analyses, the images were merged with mouse brain atlases to identify ROIs. For analysis of DAPI signals, images were converted to binary, and each signal was separated using "watershed". Signals inside the ROI were measured using "analyze particles". c-Fos signals were manually counted using the ImageJ Cell Counter plugin.

## Chemokine arrays

Frozen hindbrains were mechanically pulverized in the RIPA buffer and protease inhibitor (x100) mixture and centrifuged at 12000 rpm for 20 min at 4 °C. The supernatant was collected, and the total protein levels in the hindbrain lysates were quantified by BCA protein assay according to the manufacturer's standard protocol. Based on the protein concentration, the same amount of lysate was used for the chemokine array kit (R&D Systems, Minneapolis, MN) and conducted following the manufacturer's standard protocol. The chemokine array signals were visualized using the FluorChem™ R (ProteinSimple, San Jose, Ca) after 20 min X-ray exposure. The intensity of signals was measured and analyzed by Image J software. The average value from duplicate spots was achieved, and the chemokine expression was calculated as a relative expression of control gp130 expression, and then calculated as the ratio of values in CnH99 alone infected hindbrain to the naive hindbrain.

## RNA extraction and RT-qPCR analyses

Total RNA was extracted from CD4+ T cells using TRIsure reagent (Bioline USA Inc. USA) according to the manufacturer's standard protocol. cDNA synthesis was performed with qScript cDNA SuperMix (Quanta). qPCR analysis was performed with KAPA-SYBR-FAST (KAPA BioSystems) and QuantStudio 3 Real-Time PCR System (Applied Biosystems) with an initial denaturing step at 95 °C for 2 min, followed by 40 cycles of a denaturation step (95 °C for 10 s) and an annealing/extension step (60 °C for 30 s). Gene expression was determined using the relative quantification PCR double delta cycle threshold (ΔΔCt) method with $Actb$ (β-actin) as an internal control. Primers are listed in Supplementary Table 1.

## Flow cytometry

Seven days after CD4+ T cell transfer, brains and lungs were retrieved. Tissues were excised and minced in PBS supplemented with collagenase D (1 mg/ml). Minced tissues were incubated for 30 min at 37 °C, filtered through the 80-μm mesh, and then centrifuged at $277 \times g$ at 4 °C. To isolate mononuclear cells from the brains and lungs, cells were resuspended in 30% Percoll (in PBS), laid over 70 % Percoll, and centrifuged at $377 \times g$ for 20 min at room temperature. Before cell staining with antibodies, Fc receptors were blocked with Fc Block (BD Pharmingen) for 7 min on ice. Cells were stained with fluorochrome-conjugated specific antibodies (1:200) for 30 min on ice, followed by washing and collecting cells. Stained cells were analyzed on Cytek Aurora with the FCS Express (De Novo). Antibodies used were: APC-Fire 750 CD45 (Cat# 103154, BioLegend), PE/Cy5 CD3 (Ref# 15−0031, eBioscience), PE/Cy7 CD4 (Cat# 100422, BioLegend), FITC CD4 (Cat# 100406, BioLegend), PE/Cy7 CD19 (Cat# 115520, BioLegend), PE/Cy7 CD11c (Cat# 117318, BioLegend), Alexa647 F4/80 (Lot# 1862665, Invitrogen), FITC Ly6C (Cat# 128006, BioLegend), Pacific Blue Ly6G (Cat# 127612, BioLegend), PE CD11b (Cat# 101208, BioLegend), Alexa647 Sema6b (sc-390928, Santa Cruz Technology), and Alexa 480 Ephrin B3 (sc-390696, Santa Cruz Technology). Data were analyzed using FCS Express version 6. Acccording to previous study[13], we identified immune cell population in the lungs and the brains. In the lung samples, DC or interstitial or exudate macrophages (DC/iMΦ/eMΦ) were identified as CD45+CD11c+F4/80-, alveolar macrophages (AM) as CD45+CD11c+F4/80+, inflammatory macrophage/monocyte as CD45+Ly6C+CD11b+, neutrophils as CD45+Ly6GhiCD11b+, and CD4+ T cells as CD45+CD3+CD4+. In the brain samples, CD4+ T cells were identified as CD45+CD3+CD4+, microglia as CD45loF4/80+, macrophage as CD45hiF4/80+, and neutrophils as CD45+Ly6G+.

## Material

Mouse ephrin B3 (sc-39441-V), semaphorin 6B (sc-63007-V), and control shRNAs (sc-108080) were purchased from Santa Cruz Biotechnology. Maraviroc was purchased from VWR (101760-358).

## Statistics

A sample size of n ≥ 3 was used for all statistical analyses. Statistical analysis was evaluated with two-tailed unpaired Student's $t$ tests, Tukey's multiple comparison test, or Log-rank (Mantel−Cox) test following one-way ANOVA using GraphPad Prism Version 9. Data are presented as mean ± SEM, and $p$ values *$p \le 0.05$, **$p < 0.01$, and ***$p < 0.001$ were considered statistically significant. Animals were randomly used for experiments under the criteria aforementioned in the section "Animals." All behavior experiments were performed in a blinded and randomized fashion. No animals or data points were excluded. Experiments were repeated at least two times. Measurements were taken from distinct samples for each experiment. For each test, the experimental unit was an individual animal. No statistical methods were used to predetermine sample sizes, but our sample sizes are similar to those generally employed in the field[13]. Data distribution

was assumed to be normal, but this was not formally tested. Statistical analyses and graphical presentations were computed with GraphPad Prism software (GraphPad, La Jolla, United States). Where appropriate, the n (number of mice or biological samples) and $p$ values are indicated in the figure legends.

### Reporting summary
Further information on research design is available in the Nature Portfolio Reporting Summary linked to this article.

## Data availability
The raw data in this study are available. All data are provided in the Source data file. The atlas used was the Mouse Brain in Stereotaxic Coordinates by Keith B.J. Franklin and George Paxinos. Source data are provided with this paper.

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

## Acknowledgements

We thank Mary Clutter, Chris Oh, Eunjoo Kang, Dr. Yee Ming Khaw, and Joshua Saylor for helping with samples isolation and analysis. We acknowledge that mCherry-CnH99 were provided by Dr. Andrew Alspaugh (Duke University). We acknowledge that pAAV-hSyn-DIO-hMeD(Gq)-mCherry were provided by Dr. Bryan Roth (The University of North Carolina at Chapel Hill). This research was supported by NIH R01-AI136999 (M.I.).

## Author contributions

T.K., J.Z., S.A., and M.I. planned the experiments and wrote the manuscript. T.T. discussed the experiments with M.I. T.K., S.A., and M.I. performed sample isolation and behavior study. T.K. performed brain histology and chemokine assay. J.Z. and T.K. performed the N2a cell co-culture study. T.K. and Y.I. performed the brain neuronal network study. J.Z. and A.H.D. performed CRSR analysis. S.A. performed qPCR analysis. J.Z. and K.B. performed fungal burden studies and tissue processing. H.S. conducted tissue processing. J.Z. and S.A. performed flow

cytometry. S.A. and M.I. performed pulmonary function tests. K.S. discussed the WBP experiments with S.A. and M.I., instructed S.A. and M.I. about WBP equipment, and edited the manuscript.

## Competing interests

The authors declare no competing interests.
