## [Peer Review File · Nature Communications]

T cell infiltration into the brain triggers pulmonary dysfunction in murine *Cryptococcus*-associated IRISREVIEWER COMMENTS

Reviewer #1 (Remarks to the Author):

The paper from an accomplished neuroscience group describes pulmonary effects linked to T-cell-driven neuronal damage in nucleus tractus solitarius in cryptococcus-associated immune reconstitution inflammatory syndrome model.

Exciting and novel are the results from high-quality brain imaging studies, that show the precise localization of CNS damage, neuronal network disconnection. The explorations of pathways of neuronal damage, the studies supporting the roles of semaphorin 6B and ephrin B3 in the activation of neuro-pathogenic IRIS T-cells are very interesting.

The weaknesses are an underdeveloped brain immunology part and a lack of other parameters in addition to the respiratory rate to define pulmonary dysfunction. Thus some conclusions in this study are not fully solidified, at this time.

Specific Comments

1. Ln 114-115. This conclusion that "B cells are not involved in C-IRIS induction with CD4+ T cell transfer" makes not too much sense immunologically. To clarify this should not the authors transfer the T-cells, B-cells, or combination of both T and B-cells to the CnH99-Rag1^{-/-} mice and directly address this point? The data shown here only demonstrate that in the Tcra^{-/-} mice (which make B-cells) transfer of T-cells is sufficient to induce IRIS.

2. Ln 118-119. The conclusion "that mortality under the C-IRIS condition may be caused by pulmonary dysfunction" should be solidified by more direct evidence. Did the mice with IRIS demonstrate hypoxia and or hypercapnia? Were pathological breathing patterns observed besides reduced respiratory frequency in C-IRIS mice? In view of this reviewer, a decrease in respiratory rate is not sufficient to define pulmonary dysfunction.

3. In all presented models, what were the fungal burdens in the brain? Could any of the observed phenotypes be associated with higher and lower brain CFU?

4. The data on CCL8-CCR5 axis appear incomplete. It would be important in addition to the Maraviroc study if CCR5KO mice T-cells transferred iv of the CnH99 Tcra^{-/-} mice would better address the role of CCR5 axis in T-cell infiltration of the brain and their role in the neuro/respiratory phenotype.

5. Were experiments repeated on more than 1 occasion to ensure reproducibility?

Minor:

Some conclusions in the discussion are unclear.

Ln 229-31 It is unclear which wild-type mice data the authors refer to in comparison to Tcra^{-/-} mice. There seems to be no comparison between WT and Tcra mice.

Ln233-35 The results with strain 52D were discouraging, so it is unclear why authors suggest using it experimentally to induce C-IRIS?

Ln 241 and Ln 254-5. Was the pulmonary dysfunction here defined using the same or different criteria? No reference is given for the statement on Ln 241. The conclusion in Ln 254-5 is speculative

LN 267-9. The claim that the mechanism of CD4+ TC infiltration was defined, is not sufficiently substantiated by the data presented. (As stated in comment 4). The inhibitor data are limited and insufficient to make definitive conclusions. Results are puzzling considering that the T-cells were injected into the CNS already. How was the infiltration defined?

Reviewer #2 (Remarks to the Author):

Reviewer (Remarks to the Author):

This manuscript describes a potential mechanism for pulmonary dysfunction in a mouse model of Cryptococcus-associated immune reconstitution inflammatory syndrome (C-IRIS) whereby pathogenic CCR5+ CD4+ T cells infiltration into the CNS leads to neuronal damage in the nucleus tractus solitarius (NTS) leading to respiratory dysfunction. Given these findings, the authors go on to present data supporting the efficacy of the antiretroviral drug, Maraviroc, in mitigating signs of pulmonary dysfunction in mice.

In general, the findings potentially broaden our understanding of pulmonary dysfunction following C-IRIS using a mouse model and would be of interest not only to fungal immunologists but also neurobiologists. However, 1) the rationale and contexts from human pathophysiology is overstated, 2) many aspects including the pulmonary physiology are only superficially studied and 3) the model rests entirely on a phenotype based on T-cells, ignoring potentially contributing or antagonizing effects from myeloid and non-hematopoietic cells.

Major Critiques:

Rationale: The authors strongly state that their IRIS model simulates that of the human condition which has little if no evidence at the current time. Specifically,

L49: C-IRIS is a common manifestation of cryptococcal meningoencephalitis after initiation of anti-retroviral therapy, not the other way around.

L60: The HAART designation is obsolete, now anti-retroviral therapy is referred to as ART.

L65: Pulmonary disease only rarely results in clinical pathology compared to that of CNS. Pts require therapy for pulmonary disease cIRIS only rarely (usually tracheal obstruction from lymph nodes) and recurrence of cryptococcal disease after steroids is unusually rare and most likely due to non-compliance on the part of the patient. It is certainly inaccurate to say that recurrences are common.

L96: condition "in mice" is required in the text as neuronal-pulmonary dysfunction in humans has not been demonstrated. One could say that it establishes the capacity for this mechanism, but it could be specific to mice and does not suggest the mechanism in humans. It certainly provides an interesting mechanism for further study.

L105: "because pulmonary disease is a primary symptom in C-IRIS..." This statement is inaccurate and unnecessary to justify the study. The primary symptoms in C-IRIS are neurological and consist of lethargy, headaches and cranial nerve abnormalities. Any reference to similar findings of slower respiratory rates should be the primary studies for the conclusion.

L119: would rephrase to something like 'could contribute to...'

L227: Recently, we established a clinically relevant model.." This remark should be removed as any clinical correlation has not been established as yet. It is an interesting model and demonstrates the potential for such a mechanism, but a finding in a single strain of mice induced by adoptive transfer of lymphocytes from a single strain of cryptococcus can hardly be described as clinically relevant at this time. All such clinical references are premature at this point

L240: "It has been reported that bilateral CNS lesions lead to neurogenic respiratory failure" No reference was given here.

Research Placed in Context with prior study (Khaw et al., PMID: 33133067): The prior study should be adequately summarized to give context for the novel aspects of this manuscript for those unfamiliar with the C-IRIS mouse model. It is somewhat unclear how this study fits in with the prior study and why both Rag1^{-/-} and Tcra^{-/-} mice are used to understand pulmonary dysfunction, while only Rag1^{-/-} mice were studied previously.

Pulmonary Studies:

L108: The authors conduct extensive neuronal studies but the respiratory studies are quite superficial. For example, respiratory rate decrease is only 1 measure of respiratory capacity. Peak volume, negative inspiratory pressure and flow related parameters would be important to implicate

a neuro-pulmonary physiology. In addition, lower respiratory rate could be driven by a relative ankylosis in the animals without participation of a neuro-pulmonary axis which can easily be measured by simple blood gas measurements of animals.

Adoptive Transfer experiments

Rationale: The model jumps to a lymphocyte adoptive transfer experimental design without testing whether the infected intact animal develops the same symptoms or pathology. This thus removes possible contributions or antagonisms from other possible hematopoietic (or non-hematopoietic) cell populations. This further compromises the integrity of the model. In addition, the findings suggest an antigen specific effect in that only their strain of H99 and not 52D results in the findings, though there is no evidence to support or characterize this phenomenon. These results could be interesting but have not been pursued. Lack of such characterization makes it difficult to put the present results in context with other studies. For example:

L136: The authors find that T-cell adoptive transfer experiments only from lymphocytes of mice infected with strain H99 and not 52D achieve changes in respiratory rate. The authors then conclude in the present manuscript, "The function of CD4+T cells in the brain is essential for the induction of pulmonary dysfunction and mortality..." However, 52D is known to result in destructive neuro inflammation that results in significant neuropathology and weight loss and is blocked by antibody to CD4. More work is thus required to understand this apparent incongruity as it questions the validity of the conclusions.

Experimental Details

Description of the adoptive transfer experiments needs clarification in text and in the figure: The description of the adoptive transfer experiment is very confusing to follow and the diagram in Figure 1D appears incomplete. More detail should be added in order to accurately indicate the following: genotype of all mice used to harvest CD4 T cells for adoptive transfer and mice that received the transfer, whether mice were infected or naïve prior to isolation of CD4 T cells and at each stage in the experiment, organ of origin of CD4 T cells harvested, and route of adoptive transfer and infection. From the text it seems like there are 3 mice involved in each experiment from pre-infection of CD4 T cell donor mice to the analysis of survival in final recipient mice, which doesn't seem possible from the diagram and typical application of adoptive transfer experiments. A detailed experimental timeline would help for Figure 1 and Figure 2.

Additional experiments will be required to support adoptive transfer experiments:

- 1) Confirmation that the CD4 T cells transferred to the recipient mouse in these experiments reach the brain and/or lung. Do cell frequencies change in the lung following transfer? How does the transfer of CD4 T cells alter B cell numbers in the brain and lung of *Tcra*^{-/-} recipient mice.
- 2) For the adoptive transfer of *Sema6b* and *Efnb3* shRNA KD experiments in Figure 4, please include qRT-PCR and western/flow confirmation of successful knockdown in CD4 T cells.
- 3) To complement studies using the CCR5 antagonist, Maraviroc, adoptive transfer of T cells from CCR5 KO mice into *Tcra*^{-/-} mice would be informative. Alternatively, i.p. injection of blocking antibody to CCL8 might strengthen the mechanistic studies. Additionally, one would expect that gene expression of *Sema6b* and *Efnb3* in CD4 T cells from Maraviroc-treated mice would be unchanged compared to untreated mice. In this case, reduced presence of pathogenic CCR5 T cells would be responsible for neuronal damage. For this reason, analysis of *Sema6b* and *Efnb3* gene expression in CD4 T cells from Maraviroc-treated mice would help to rule out a direct effect of CCR5 inhibition on CD4 T cell expression of these inhibitory molecules.

Statistically speaking, were these experiments performed twice with data from a single replicate plotted in the figures? It is unclear based on the figure legends. Please include this information.

Minor Critiques:

This manuscript should be carefully edited for clarity and precision of language. At times, the logic behind arguments made in the manuscript were difficult to follow or completely absent. Particularly in connection to introducing the connection between CNS disease to pulmonary dysfunction (Introduction Line 71-74 and 84-91). Also, since this manuscript covers research of interest to many groups (fungal immunologists, neurologist, clinicians, etc.) it would be helpful to

provide a bit more background into the rationale for techniques chosen to address experimental questions. For example, it would be helpful to your diverse audience to mention how c-Fos is well known to be expressed following depolarization of neurons before describing c-Fos staining results in order to give adequate context for its application.

Line 50: Doesn't CM precede C-IRIS and not the other way around? Please correct this statement in text.

Line 65: How common is pulmonary disease presentation in C-IRIS? Clinical relevance would be important details to highlight. Along these lines, it would be informative to connect details related to the CCL8-CCR5 axis identified in mice to published clinical data seen in C-IRIS patients (for example, incorporating what is known from analysis of patient CSF samples identifying CD4 T cells phenotypes or chemokine expression in the CSF).

Line 114: "Tcra^{-/-} mice with C-IRIS treatment also cause mortality." Was mortality not seen in the Rag1^{-/-} mice infected with CnH99? According to Khaw et al (PMID: 33133067), mortality was seen in these mice. When comparing Tcra^{-/-} C-IRIS infection data to Rag1^{-/-} C-IRIS infection data, please indicate in text relevant results from Khaw et al.

Supplementary Figure 2: Minor points. Is the Cn strain used in this figure Cn 52D? Please correct the species number. Also, in Line 116: You show that pulmonary dysfunction is a Cn H99 (Serotype A)-associated IRIS specific phenotype and that this matched the mortality phenotype. Would pulmonary dysfunction present in mice if the dose of Cn 52D was increased to a dose that matched the mortality difference seen with Cn H99?

Supplementary Figure 3: Were all these brain slices sequentially sliced and from a single mouse at 7 dpt? It is not clear from the legend.

L118-119: awkward, should be rewritten

Line 151: The citation field code seems to have been lost for this reference.

Supplementary Figure 4: How many sections were quantified from each mouse?

Line 211-12: Where T cells adoptively transferred in these experiments?

Line 214: Figure 5B In what tissue (the brain?) was this analysis performed?

Figure 5E: Were 8 mice per group analyzed with each dot representing a single mouse?

L 232: strain Cn52. Is this 52D? The authors should use ATCC designations wherever possible to exclude the possibility of differences in strains leading to ambiguities and loss of reproducibility.

Line 234: Should this sentence read: "Therefore, we strongly suggest against using Cn serotype D strain. . . "?

L304: yeast strains: ATCC and catalog numbers should be used and fresh isolates without serial plating should be used and documented so as to ensure reproducible results.

Reviewer #3 (Remarks to the Author):

This paper investigates the mechanism by which respiratory rate is slowed in a mouse model of cryptococcal meningitis. This model, published previously, described infecting T- and B -cell deficient mice with a low inoculum of *C. neoformans* and, after three weeks gave the mice an intravenous injection of T cells from wild type mice. Over the next week the T-cell reconstituted mice lost weight, began to die and had slowed respiration. As published before, the reconstituted mice did not have more cryptococci in their brain or lungs than unreconstituted controls but had more T cells in their lungs and brain.

The current study uses T-cell deficient mice and uses a variety of procedures to show that infected, reconstituted mice have neuronal damage to their posterolateral medulla. Neuronal damage was assessed by loss of neurons on histology and decreased neurites extending out of neurons. T-cells needed to cause neuronal damage require expression of certain ephrin and semaphoring genes, as judged by gene silencing. Neuronal damage in vitro could be blocked by maraviroc.

This manuscript would benefit by several changes. Foremost, the term, "pulmonary dysfunction" should be changed to "respiratory rate depression" to clarify what is meant. Also important is to convince the reader that respiratory rate depression is an essential feature in the pathogenesis of disease in their model. The significance of this work needs better validation. The argument (line 105) is not correct that respiratory depression is a critical component of the immune reconstitution syndrome in antiretroviral treated HIV patients with cryptococcosis. The central nervous system manifestations are headache, seizures and increased opening pressure on lumbar puncture. Also, caution is needed for the conclusion that T-cells in the nucleus tractus solitarius cause the slower respiration. There are viable cryptococci in the brain that also damage the CNS and damage by T-cells is not confined to that one neural pathway.

Considering that the key phenotype being assessed is respiratory rate, it should be clearer how this was measured. A device on a forepaw would not be adequate (lines 329-330). Whole body plethysmography would be preferred. It should be stated whether the mice were sedated to measure respiratory rate, and if so, how.

Some of the figures are too dark to read: 2D, 3, 4D and 5D. It is unclear what the reader should see in Fig. 1C and supp. Fig 3. If Fig 1C is intended to show brain edema, brain weights are a better measure. In supp Fig 2B and 2C, I cannot tell if the single line represents both groups. The little squares may also have circles alongside them.

There may be some translation errors. What does "signature" on line 73 mean? The sentence on line 119 needs to be unscrambled. Reference 27 on line 271 is wrong. On lines 233-235 don't you mean "not using" instead of "using" strain 52?

It is odd that no stains of brains for cryptococci are mentioned. The usual histologic feature in mice and humans are clusters of cryptococci, which are the "soap-bubble cysts" mentioned on line 240. Are the black holes in the brain in Fig. supp.3 masses of cryptococci?

Detailed point by point Response to the referees' comments on the manuscript NCOMMS-22-42243 "T Cell Infiltration into the Brain Triggers Pulmonary Dysfunction in Murine Cryptococcus-associated IRIS"

Below are point-by-point responses to the reviewer's comments

Reviewer #1

We thank reviewer #1 for the time spent reading and revising our manuscript and for giving us many valuable comments to improve our manuscript. We have carefully considered all comments when revising our manuscript.

General comments:

The paper from an accomplished neuroscience group describes pulmonary effects linked to T-cell-driven neuronal damage in nucleus tractus solitarius in cryptococcus-associated immune reconstitution inflammatory syndrome model. Exciting and novel are the results from high-quality brain imaging studies that show the precise localization of CNS damage, neuronal network disconnection. The explorations of pathways of neuronal damage, and the studies supporting the roles of semaphorin 6B and ephrin B3 in the activation of neuro-pathogenic IRIS T-cells are very interesting. The weaknesses are an underdeveloped brain immunology part and a lack of other parameters in addition to the respiratory rate to define pulmonary dysfunction. Thus some conclusions in this study are not fully solidified, at this time.

We thank the referee for pointing out these comments. To address these comments, we conducted additional experiments, including immune cell profiling in the brain and lung using flow cytometry, the fungal burden in the brain and lungs, and testing more respiratory parameters using whole-body plethysmography (WBP) to define respiratory dysfunction. The results are mentioned below and in the revised manuscript. We believe the revised manuscript is substantially improved after addressing your comments.

COMMENT 1:

Ln 114-115. This conclusion that "B cells are not involved in C-IRIS induction with CD4+ T cell transfer" makes not too much sense immunologically. To clarify this should not the authors transfer the T-cells, B-cells, or combination of both T and B-cells to the CnH99-Rag1^{-/-} mice and directly address this point? The data shown here only demonstrate that in the Tcra^{-/-} mice (which make B-cells) transfer of T-cells is sufficient to induce IRIS.

RESPONSE 1:

We concur with the reviewer's comment. We totally agree with the reviewer's concern. The original text referred to the involvement of B cells, but it is impossible to discuss conclusive evidence of the involvement of B cells by comparing the data using different strains (Rag1^{-/-} and Tcra^{-/-}). Since this study focuses mainly on T cells, this section was deleted in the revised manuscript.

COMMENT 2:

Ln 118-119. The conclusion "that mortality under the C-IRIS condition may be caused by pulmonary dysfunction" should be solidified by more direct evidence. Did the mice with IRIS demonstrate hypoxia and or hypercapnia? Were pathological breathing patterns observed besides reduced respiratory frequency in C-IRIS mice? In view of this reviewer, a decrease in respiratory rate is not sufficient to define pulmonary dysfunction.

RESPONSE 2:

We agree with the reviewer that a decreased respiratory rate is insufficient to define pulmonary dysfunction. Therefore, we performed additional experiments to demonstrate pulmonary dysfunction. We measured oxygen saturation levels to evaluate hypoxia and conducted WBP to evaluate pathological breathing patterns. We found that C-IRIS mice have hypoxia and, in addition to reduced respiratory frequency, have more abnormal parameters (e.g., expiration time). We modified the manuscript to include details on these experiments in the revised manuscript (Lines 113-115; 121-126. Figure 1B, C).

COMMENT 3:

In all presented models, what were the fungal burdens in the brain? Could any of the observed phenotypes be associated with higher and lower brain CFU?

RESPONSE 3:

To address the reviewer's questions, we performed additional experiments to confirm the fungal burden in the brains of Tcra^{-/-} mice. We found a significantly higher fungal burden in the brains of C-IRIS mice (Lines 143-144, Figure 2C). The relationship between higher brain CFU and C-IRIS phenotype is an interesting topic; therefore, we discussed possible future directions in the revised manuscript (Lines 307-309 of the revised manuscript).

COMMENT 4:

The data on CCL8-CCR5 axis appear incomplete. It would be important in addition to the Maraviroc study if CCR5KO mice T-cells transferred iv of the CnH99 Tcra^{-/-} mice would better address the role of CCR5 axis in T-cell infiltration of the brain and their role in the neuro/respiratory phenotype.

RESPONSE 4:

We thank the reviewer for this comment. As the reviewer suggested, we isolated CD4⁺ T cells from CCR5KO mice, and T-cells were transferred i.v. to the CnH99-infected Tcra^{-/-} mice (C-IRIS). We found a significant decrease in T cell infiltration into the brain and reduced respiratory dysfunctions and neurite damage in the brain, compared with wild-type CD4⁺T cell-mediated C-IRIS. We added these data in the revised manuscript (Lines 247-253; Figure 7C-E in the revised manuscript).

COMMENT 5:

Were experiments repeated on more than 1 occasion to ensure reproducibility?

RESPONSE 5:

Thank you for pointing out this in the comment. In all experiments, the data were shown from experiments repeated at least two times. We added this information in the revised manuscript (Line 547).

COMMENT 6:

Some conclusions in the discussion are unclear. Ln 229-31 It is unclear which wild-type mice data the authors refer to in comparison to Tcra^{-/-} mice. There seems to be no comparison between WT and Tcra mice.

RESPONSE 6:

We thank the reviewer for the comment. We revised this sentence as follows: "we demonstrated that Tcra^{-/-} mice that received CnH99 (100 yeasts, i.n.) pre-infection and CD4⁺ T cell transfer (C-IRIS condition) also had a significantly higher mortality rate than that of control mice (naïve, CD4⁺ T cell transfer alone, CnH99 infection alone). (Lines 275-279).

COMMENT 7:

Ln233-35 The results with strain 52D were discouraging, so it is unclear why authors suggest using it experimentally to induce C-IRIS?

RESPONSE 7:

We apologize for the error in the description on lines 233-35 of the original text. We changed the text to the following: "We suggest using CnH99 in experimental C-IRIS induction to recapitulate the C-IRIS condition" in the revised manuscript (Lines 282-284).

COMMENT 8:

Ln 241 and Ln 254-5. Was the pulmonary dysfunction here defined using the same or different criteria? No reference is given for the statement on ln 241. The conclusion in Ln 254-5 is speculative.

RESPONSE 8:

We appreciate the reviewer's instructive suggestion. We have added a reference (reference¹ in this document, Line 85 in the revised manuscript) for the statement on line 241 (original) and updated the text on lines 254-5 (original) to the following: "damage to respiration-controlled neuronal functions in the NTS may be a potential mechanism for pulmonary dysfunction under the C-IRIS condition in mice". (Lines 325-326).

COMMENT 9:

LN 267-9. The claim that the mechanism of CD4+ TC infiltration was defined, is not sufficiently substantiated by the data presented. (As stated in comment 4). The inhibitor data are limited and insufficient to make definitive conclusions. Results are puzzling considering that the T-cells were injected into the CNS already. How was the infiltration defined?

RESPONSE 9:

We thank the reviewer for these valuable comments. As mentioned in our response to Comment 4, we performed additional experiments using CCR5^{-/-} mice. In both the original and revised experiments, we transferred T cells by i.v. injections to induce C-IRIS and measured how many T cells are in the brain by flow cytometry (Figures 7).

Reviewer #2

We thank the reviewer for giving us many valuable comments that dramatically improved our manuscript. We carefully considered all comments.

REVIEWER GENERAL COMMENTS:

This manuscript describes a potential mechanism for pulmonary dysfunction in a mouse model of Cryptococcus-associated immune reconstitution inflammatory syndrome (C-IRIS) whereby pathogenic CCR5+ CD4+ T cells infiltration into the CNS leads to neuronal damage in the nucleus tractus solitarius (NTS) leading to respiratory dysfunction. Given these findings, the authors go on to present data supporting the efficacy of the antiretroviral drug, Maraviroc, in mitigating signs of pulmonary dysfunction in mice.

In general, the findings potentially broaden our understanding of pulmonary dysfunction following C-IRIS using a mouse model and would be of interest not only to fungal immunologists but also neurobiologists.

1) the rationale and contexts from human pathophysiology is overstated,

2) many aspects including the pulmonary physiology are only superficially studied and

3) the model rests entirely on a phenotype based on T-cells, ignoring potentially contributing or antagonizing effects from myeloid and non-hematopoietic cells.

We thank the reviewer for the positive comments. According to the reviewer's suggestion, we modified the manuscript to more accurately describe animal models of C-IRIS and their relevance to clinical conditions. To address the reviewer's concern, we conducted experiments, including whole-body plethysmography (WBP) and flow cytometry. Details are shown below. We believe the manuscript is substantially improved after making the suggested edits.

COMMENT 1:

Rationale: The authors strongly state that their IRIS model simulates that of the human condition which has little if no evidence at the current time. Specifically,

L49: C-IRIS is a common manifestation of cryptococcal meningoencephalitis after initiation of anti-retroviral therapy, not the other way around.

L60: The HAART designation is obsolete, now anti-retroviral therapy is referred to as ART.

L65: Pulmonary disease only rarely results in clinical pathology compared to that of CNS. Pts require therapy for pulmonary disease cIRIS only rarely (usually tracheal obstruction from lymph nodes) and recurrence of cryptococcal disease after steroids is unusually rare and most likely due to non-compliance on the part of the patient. It is certainly inaccurate to say that recurrences are common.

L96: condition "in mice" is required in the text as neuronal-pulmonary dysfunction in humans has not been demonstrated. One could say that it establishes the capacity for this mechanism, but it could be specific to mice and does not suggest the mechanism in humans. It certainly provides an interesting mechanism for further study.

L105: "because pulmonary disease is a primary symptom in C-IRIS..." This statement is inaccurate and unnecessary to justify the study. The primary symptoms in C-IRIS are neurological and consist of lethargy, headaches and cranial nerve abnormalities. Any reference to similar findings of slower respiratory rates should be the primary studies for the conclusion.

L119: would rephrase to something like "could contribute to..."

L227: Recently, we established a clinically relevant model.." This remark should be removed as any clinical correlation has not been established as yet. It is an interesting model and demonstrates the potential for such a mechanism, but a finding in a single strain of mice induced by adoptive transfer of lymphocytes from a single strain of cryptococcus can hardly be described as clinically relevant at this time. All such clinical references are premature at this point

L240: "It has been reported that bilateral CNS lesions lead to neurogenic respiratory failure" No reference was given here.

RESPONSE 1:

We thank the reviewer for the comments, and we agree with the reviewer's concerns. According to the reviewer's suggestion, we changed our language thoroughly to more accurately describe the animal model and the clinical conditions (Lines 49-51, 56, 62-63, 85, 96, 105, 130, and 269-270). We also corrected the text and added references for line 240 (original) in the original manuscript (Line 85 in the revised manuscript).

COMMENT 2:

Research Placed in Context with prior study (Khaw et al., PMID: 33133067): The prior study should be adequately summarized to give context for the novel aspects of this manuscript for those unfamiliar with the C-IRIS mouse model. It is somewhat unclear how this study fits in with the prior study and why both Rag1^{-/-} and Tcra^{-/-} mice are used to understand pulmonary dysfunction, while only Rag1^{-/-} mice were studied previously.

RESPONSE 2:

We thank the reviewer for these excellent suggestions. We have added text to summarize our prior study in the introduction and discussion (Lines 68-80, 269-273). Human HIV infection primarily infects and impairs T-cell function. B cells remain in HIV patients. After our first publication, we thought that Tcra^{-/-} mice were better suited to mimic this situation. Thus, in the present study, we mainly used Tcra^{-/-} mice in the present study. Results in Tcra^{-/-} and Rag1^{-/-} mice were very similar. We discussed this issue in the revised manuscript (Lines 93-95 and 110-111).

COMMENT 3:

Pulmonary Studies:

L108: The authors conduct extensive neuronal studies but the respiratory studies are quite superficial. For example, respiratory rate decrease is only 1 measure of respiratory capacity. Peak volume, negative inspiratory pressure and flow related parameters would be important to implicate a neuro-pulmonary physiology. In addition, lower respiratory rate could be driven by a relative ankylosis in the animals without participation of a neuro-pulmonary axis which can easily be measured by simple blood gas measurements of animals.

RESPONSE 3:

We thank the reviewer for the constructive comments. We performed additional experiments to demonstrate pulmonary dysfunction. We conducted WBP to evaluate pathological breathing patterns. We found that C-IRIS mice have more abnormal parameters (e.g., expiration time) than control mice. We also measured SpO₂ and found a significant decrease in SpO₂ in C-IRIS mice, therefore, decreasing the possibility of ankylosis. We discussed the possibility of ankylosis in the discussion and updated the manuscript to include details on these experiments in the revised manuscript (Lines 113-115, 121-126, and 288-291).

COMMENT 4:*Adoptive Transfer experiments*

Rationale: The model jumps to a lymphocyte adoptive transfer experimental design without testing whether the infected intact animal develops the same symptoms or pathology. This thus removes possible contributions or antagonisms from other possible hematopoietic (or non-hematopoietic cell populations). This further compromises the integrity of the model.

RESPONSE 4:

We appreciate your valuable comment. T cell reconstitution is required to mimic the IRIS condition, which usually happens after immune reconstitution under pathogen-pre-infected conditions; therefore, we use infected *Tcra*^{-/-} mice as recipients (immunocompromised) and transfer T cells as immune reconstitution. As the reviewer pointed out, we agree that hematopoietic or non-hematopoietic cells may contribute to T cell property changes (e.g., CCR5 and neurotoxic molecules via innate immune cells) and T-cell-mediated neurodegeneration (e.g., via microglia, astrocyte, and infiltrated peripheral immune cells). We discussed this critical issue, which will be elucidated in future studies (Lines 309-312).

COMMENT 5:

In addition, the findings suggest an antigen specific effect in that only their strain of H99 and not 52D results in the findings, though there is no evidence to support or characterize this phenomenon. These results could be interesting but have not been pursued. Lack of such characterization makes it difficult to put the present results in context with other studies. For example:

L136: The authors find that T-cell adoptive transfer experiments only from lymphocytes of mice infected with strain H99 and not 52D achieve changes in respiratory rate. The authors then conclude in the present manuscript, "The function of CD4⁺T cells in the brain is essential for the induction of pulmonary dysfunction and mortality...." However, 52D is known to result in destructive neuro inflammation that results in significant neuropathology and weight loss and is blocked by antibody to CD4. More work is thus required to understand this apparent incongruity as it questions the validity of the conclusions.

RESPONSE 5:

We thank the reviewer for the comment. The strain H99 is more virulent than 52D, and in current experiments, the same amount (100 yeasts) of H99 and 52D was used as pre-infection. The lack of C-IRIS phenomenon in the case of strain 52D may be attributed to dosage, and we added discussion on this critical matter in the manuscript (Line 282-284). Because the main purpose of the present study is to elucidate key signals and molecules in C-IRIS disease, comparisons between the two strains in a dose-dependent manner will be examined in future studies.

Because of the lack of C-IRIS phenomenon with strain 52D, we did i.c.v. adoptive transfer experiments (Figure 3) using only T cells isolated from H99-infected C-IRIS mice. No significant pulmonary dysfunctions or neuronal damage were observed in the control cohorts (H99 alone and T cell alone), suggesting T cell property changes under strain H99 may be important in C-IRIS induction. I.c.v. studies demonstrated that T cells, which change their properties in the periphery in the brain play a role in pulmonary dysfunctions and mortality directly or indirectly via interactions with cryptococcus or other immune cells. We have modified the original statement in the revised manuscript (Lines 165-167).

COMMENT 6:

Experimental Details

Description of the adoptive transfer experiments needs clarification in text and in the figure:

The description of the adoptive transfer experiment is very confusing to follow and the diagram in Figure 1D appears incomplete. More detail should be added in order to accurately indicate the following: genotype of all mice used to harvest CD4 T cells for adoptive transfer and mice that received the transfer, whether mice were infected or naïve prior to isolation of CD4 T cells and at each stage in the experiment, organ of origin of CD4 T cells harvested, and route of adoptive transfer and infection. From the text it seems like there are 3 mice involved in each experiment from pre-infection of CD4 T cell donor mice to the analysis of survival in final recipient mice, which doesn't seem possible from the diagram and typical application of adoptive transfer experiments. A detailed experimental timeline would help for Figure 1 and Figure 2.

RESPONSE 6:

We appreciate the reviewer's valuable suggestion. As the reviewer pointed out, we added more details of the adoptive transfer experiments to the experimental scheme graph in Figure 3A and text (Lines 154-159).

COMMENT 7:

Additional experiments will be required to support adoptive transfer experiments:

1) Confirmation that the CD4 T cells transferred to the recipient mouse in these experiments reach the brain and/or lung. Do cell frequencies change in the lung following transfer? How does the transfer of CD4 T cells alter B cell numbers in the brain and lung of Tcra^{-/-} recipient mice.

RESPONSE 7:

We appreciate and agree with the reviewer's comment. To address your questions, we conducted flow cytometry in the brains and lungs and found a significant number of CD4 T cells in the brain compared to controls. We also found higher numbers of macrophages and neutrophils in the brains and higher numbers of neutrophils in the lungs but no change in B cells. We added details in the revised manuscript (Lines 145-148, Figure 2D, Supplementary Fig. 3).

COMMENT 8:

2) For the adoptive transfer of Sema6b and Efnb3 shRNA KD experiments in Figure 4, please include qRT-PCR and western/flow confirmation of successful knockdown in CD4 T cells.

RESPONSE 8:

We appreciate the reviewer's instructive suggestion. We confirmed the efficacy of *Sema6b* and *Efnb3* knockdown by quantitative RT-PCR and flow cytometry. We added details in the revised manuscript (Lines 218-219, Supplementary Fig. 5).

COMMENT 9:

3) To complement studies using the CCR5 antagonist, Maraviroc, adoptive transfer of T cells from CCR5 KO mice into Tcra^{-/-} mice would be informative. Alternatively, i.p. injection of blocking antibody to CCL8 might strengthen the mechanistic studies. Additionally, one would expect that gene expression of Sema6b and Efnb3 in CD4 T cells from Maraviroc-treated mice would be unchanged compared to untreated mice. In this case, reduced presence of pathogenic CCR5 T

*cells would be responsible for neuronal damage. For this reason, analysis of *Sema6b* and *Efnb3* gene expression in CD4 T cells from Maraviroc-treated mice would help to rule out a direct effect of CCR5 inhibition on CD4 T cell expression of these inhibitory molecules.*

RESPONSE 9:

We appreciate the reviewer's constructive and critical suggestions. We transferred CD4⁺ T cells isolated from CCR5^{-/-} mice to CnH99-preinfected Tcra^{-/-} mice to induce C-IRIS, and we found less T cell infiltration into the brain, reduced respiratory dysfunctions, and less neurite damage (Lines 247-253, Figure 7C-E). In addition, we also confirmed that *Sema6b* and *Efnb3* gene expression was unchanged by the maraviroc treatment, suggesting CCR5 signal is not involved in T cell neurotoxic property change under the C-IRIS condition (Lines 260-265, Supplementary Fig. 6).

COMMENT 10:

Statistically speaking, were these experiments performed twice with data from a single replicate plotted in the figures? It is unclear based on the figure legends. Please include this information.

RESPONSE 10:

We appreciate the reviewer's critical suggestion. We have confirmed reproducibility in independent experiments and shown the representative data. We added this information to the figure legends in the revised manuscript (Line 547).

COMMENT 11:

Minor Critiques:

*This manuscript should be carefully edited for clarity and precision of language. At times, the logic behind arguments made in the manuscript were difficult to follow or completely absent. Particularly in connection to introducing the connection between CNS disease to pulmonary dysfunction (Introduction Line 71-74 and 84-91). Also, since this manuscript covers research of interest to many groups (fungal immunologists, neurologist, clinicians, etc.) it would be helpful to provide a bit more background into the rationale for techniques chosen to address experimental questions. For example, it would be helpful to your diverse audience to mention how *c-Fos* is well known to be expressed following depolarization of neurons before describing *c-Fos* staining results in order to give adequate context for its application.*

RESPONSE 11:

We appreciate the reviewer's instructive suggestion. We have carefully and thoroughly modified the manuscript according to the comment (Line 81-91, 198-199).

COMMENT 12:

Line 50: Doesn't CM precede C-IRIS and not the other way around? Please correct this statement in text.

RESPONSE 12:

We appreciate and agree with the reviewer's comment. We revised the Introduction section to state "C-IRIS is a common manifestation of cryptococcal meningitis (CM) and is characterized by central nervous system (CNS) complications." (Lines 49-51)

COMMENT 13:

Line 65: How common is pulmonary disease presentation in C-IRIS? Clinical relevance would be important details to highlight. Along these lines, it would be informative to connect details related to the CCL8-CCR5 axis identified in mice to published clinical data seen in C-IRIS patients (for example, incorporating what is known from analysis of patient CSF samples identifying CD4 T cells phenotypes or chemokine expression in the CSF).

RESPONSE 13:

We appreciate the reviewer's instructive comments. Pulmonary disease is a recognized symptom in C-IRIS; several case reports have been published²⁻⁵, but epidemiology studies are lacking. It has been reported that *CCL8* mRNA is elevated in whole blood samples in patients who developed C-IRIS after ART treatment compared to those who did not develop C-IRIS⁶. It also has been reported that the expression of *CCL8* is elevated when heat-killed *Cryptococcus neoformans* are stimulated with PBMC for 24 h examined by RNA-seq⁷. Thus, increased production of *CCL8* may trigger C-IRIS, and it is considered clinically beneficial to validate treatments that target the *CCL8-CCR5* axis. We added these informations in the Introduction and Discussion sections of the revised manuscript (Lines 85-88 and 344-348).

COMMENT 14:

Line 114: "Tcra-/- mice with C-IRIS treatment also cause mortality." Was mortality not seen in the Rag1-/- mice infected with CnH99? According to Khaw et al (PMID: 33133067), mortality was seen in these mice. When comparing Tcra-/- C-IRIS infection data to Rag1-/- C-IRIS infection data, please indicate in text relevant results from Khaw et al.

RESPONSE 14:

We thank the reviewer for the comment. We have added relevant previous results to the revised manuscript (Lines 73-75, 128).

COMMENT 15:

Supplementary Figure 2: Minor points. Is the Cn strain used in this figure Cn 52D? Please correct the species number. Also, in Line 116: You show that pulmonary dysfunction is a Cn H99 (Serotype A)-associated IRIS specific phenotype and that this matched the mortality phenotype. Would pulmonary dysfunction present in mice if the dose of Cn 52D was increased to a dose that matched the mortality difference seen with Cn H99?

RESPONSE 15

We thank the reviewer for the comment. We have corrected the strain name. We believe the pathogenesis of 52D strain may be a dose-dependent. Therefore, pulmonary dysfunction would be present in mice if the dose of Cn52D is increased to a dose that matched the mortality difference seen with CnH99. It is an interesting topic that should be considered in our future studies (Lines 282-284).

COMMENT 16:

Supplementary Figure 3: Were all these brain slices sequentially sliced and from a single mouse at 7 dpi? It is not clear from the legend.

RESPONSE 16:

We appreciate the reviewer's critical suggestion. We added detailed information in the revised manuscript (Supplementary Fig. 2 legend).

COMMENT 17:

L118-119: awkward, should be rewritten

RESPONSE 17:

We appreciate the reviewer's comment. We revised this sentence in the revised manuscript (Line 130).

COMMENT 18:

Line 151: The citation field code seems to have been lost for this reference.

RESPONSE 18:

We appreciate the reviewer's comment. We revised this error in the revised manuscript (Lines 177-178).

COMMENT 19:

Supplementary Figure 4: How many sections were quantified from each mouse?

RESPONSE 19:

We added details to the methods section in the revised manuscript (Lines 456-457).

COMMENT 20:

Line 211-12: Where T cells adoptively transferred in these experiments?

RESPONSE 20:

We apologize for causing confusion because of a lack of detailed information on this study. Because we speculated that infiltrated CnH99 into the brain would upregulate chemokines to attract CD4⁺ T cells from the periphery, we evaluated chemokine expression in the brain without CD4⁺ T cell transfer. Because we previously detected CnH99 infiltration in the brain at four weeks post CnH99 infection, we used brain samples four weeks after CnH99 infection. We modified the sentence in the revised manuscript (Lines 237-243).

COMMENT 21

Line 214: Figure 5B In what tissue (the brain?) was this analysis performed?

RESPONSE 21:

We appreciate the reviewer's comment. We evaluated CCR5 expression in the CD4⁺ T cells isolated from lungs of C-IRIS, compared with that from CD4⁺ T cell transfer alone. We did not use brain sample for this study, because CD4⁺ T cell number is very limited in the group of CD4⁺

T cell transfer alone (control). We revised the results section (Lines 245-247) and the corresponding figure legend.

COMMENT 22:

Figure 5E: Were 8 mice per group analyzed with each dot representing a single mouse?

RESPONSE 22:

Yes. Each dot notation indicated a single mouse, meaning 8 mice/group. We explained this in the revised manuscript.

COMMENT 23:

L232: strain Cn52. Is this 52D? The authors should use ATCC designations wherever possible to exclude the possibility of differences in strains leading to ambiguities and loss of reproducibility.

RESPONSE 23:

We appreciate the reviewer's instructive suggestion. We specified the yeast strains according to ATCC designations in the methods section (Lines 385-386 in the revised manuscript). And we unified the same description in the revised manuscript.

COMMENT 24:

Line 234: Should this sentence read: "Therefore, we strongly suggest against using Cn serotype D strain. . . "?

RESPONSE 24:

We apologize for the error in the description on line 234 of the original text; we have modified it to read: "We suggest using CnH99 in experimental C-IRIS induction to recapitulate the C-IRIS condition." (Lines 282-283).

COMMENT 25:

L304: yeast strains: ATCC and catalog numbers should be used and fresh isolates without serial plating should be used and documented so as to ensure reproducible results.

RESPONSE 25:

We appreciate the reviewer's instructive suggestion. We agree with your opinion and have clarified yeast strains and handling in the methods section. (Lines 385-387 in the revised manuscript)

Reviewer #3

We thank reviewer #3 for giving us many valuable comments to improve our manuscript. We carefully considered all comments.

COMMENT 1:

Foremost, the term, “pulmonary dysfunction” should be changed to “respiratory rate depression” to clarify what is meant.

Also important is to convince the reader that respiratory rate depression is an essential feature in the pathogenesis of disease in their model. The significance of this work needs better validation.

The argument (line 105) is not correct that respiratory depression is a critical component of the immune reconstitution syndrome in antiretroviral treated HIV patients with cryptococcosis.

The central nervous system manifestations are headache, seizures and increased opening pressure on lumbar puncture.

caution is needed for the conclusion that T-cells in the nucleus tractus solitarius cause the slower respiration. There are viable cryptococci in the brain that also damage the CNS and damage by T-cells is not confined to that one neural pathway

RESPONSE 1:

We are grateful for the reviewer's crucial comments. In the revised manuscript, we performed whole body plethysmography (WBP) to examine pulmonary functions. We now use 'pulmonary dysfunction' as an umbrella term to describe all abnormal parameters from respiratory-related experiments. And we have changed to describe a decrease in respiratory rate as respiratory rate depression.

We have changed text on line 105 (original) to read: "Because pulmonary manifestations have been reported in patients²⁻⁵" (Line 105 in the revised manuscript).

In the current study, we focused on T cell functions in the *nucleus tractus solitarius* (NTS), because the NTS neurons are well-known to control respiratory functions¹. We agree with the reviewer on T cell functions in other brain regions, and we discussed this in the manuscript. We also agree that T cells in the brain may cause damage directly or indirectly via interactions with cryptococcus (e.g., location and number), which we also discussed in the revised manuscript (Lines 149-151, 305-307).

COMMENT 2:

Considering that the key phenotype being assessed is respiratory rate, it should be clearer how this was measured. A device on a forepaw would not be adequate (lines 329-330). Whole body plethysmography would be preferred. It should be stated whether the mice were sedated to measure respiratory rate, and if so, how.

RESPONSE 2:

We thank you for your constructive comments. In the original manuscript, we used the SomnoSuite device to measure respiratory rate under anesthesia by placing the device on a hind paw. In the current manuscript, we performed WBP to measure pulmonary functions under no anesthesia. We added details to the revised manuscript (Lines 121-126, Line 411).

COMMENT 3:

Some of the figures are too dark to read: 2D, 3, 4D and 5D.

RESPONSE 3:

We thank the reviewer for this suggestion. We adjusted the brightness according to the suggestion in the revised manuscript.

COMMENT 4:

It is unclear what the reader should see in Fig. 1C and supp. Fig 3. If Fig 1C is intended to show brain edema, brain weights are a better measure.

RESPONSE 4:

We thank the reviewer for this important comment. We intended to demonstrate tissue damage and cysts in the brain in the C-IRIS condition in Fig.1C and Supp. Fig 3 in the original manuscript (Figure 2 and Supplementary Fig. 2 in the revised manuscript). In the revised manuscript, we conducted experiments on Cn distribution using mCherry-CnH99 provided by Dr. Alspaugh (Duke University) and found CnH99 localization to the areas of tissue damage. We speculate that this tissue damage results from the colonization of Cn. Still, such damage is only present in C-IRIS conditions, suggesting T cells may change Cn distribution. We will add this possibility in the revised manuscript (Lines 149-151, 305-307).

Regarding brain edema, we have previously reported that C-IRIS mice showed brain edema⁸. Therefore, we did not perform such study in the present manuscript. In the revised manuscript, we better summarized our previous results (Lines 73-75).

COMMENT 5:

In supp Fig 2B and 2C, I cannot tell if the single line represents both groups. The little squares may also have circles alongside them.

RESPONSE 5:

We appreciate the reviewer's instructive suggestion. We have improved the representation of Figure styles to make it easier to be understood.

COMMENT 6:

There may be some translation errors. What does “signature” on line 73 mean? The sentence on line 119 needs be unscrambled. Reference 27 on line 271 is wrong. On lines 233-235 don't you mean “not using” instead of “using” strain 52?

RESPONSE 6:

We thank the reviewer pointed out these errors. We carefully checked all sentences and edited errors, including these sentences in the revised manuscript (Lines 68, 130, 341, 282-283).

COMMENT 7:

It is odd that no stains of brains for cryptococci are mentioned. The usual histologic feature in mice and humans are clusters of cryptococci, which are the “soap-bubble cysts” mentioned on line 240. Are the black holes in the brain in Fig. supp.3 masses of cryptococci?

RESPONSE 7:

We thank the reviewer for pointing out this critical point. In the revised manuscript, we investigated the distribution of CnH99 in the brain using mCherry-CnH99, which localizes the areas of tissue damage or cysts (Figure 2B in the revised manuscript). Thus, the black holes, detected by Ultramicroscopy in Supplementay Fig. 3 (Supplementary Fig. 2 in the revised manuscript, indicate tissue damage (multicystic encephalomalacia), in the brain. We will add this possibility in the revised manuscript (Lines 137-140).

REFERENCES

- 1 Parayil Sankaran, B., Wortman, S. B., Willemsen, M. A. & Balasubramaniam, S. Teaching NeuroImage: Bilateral Nucleus Tractus Solitarius Lesions in Neurogenic Respiratory Failure. *Neurology* **98**, e103-e104, doi:10.1212/wnl.00000000000012614 (2022).
- 2 Hu, Z. *et al.* Pulmonary cryptococcal immune reconstitution syndrome in a person living with HIV: a case report. *International journal of STD & AIDS* **31**, 280-284, doi:10.1177/0956462419893545 (2020).
- 3 Skiest, D. J., Hester, L. J. & Hardy, R. D. Cryptococcal immune reconstitution inflammatory syndrome: report of four cases in three patients and review of the literature. *The Journal of infection* **51**, e289-297, doi:10.1016/j.jinf.2005.02.031 (2005).
- 4 Haddow, L. J. *et al.* Cryptococcal immune reconstitution inflammatory syndrome in HIV-1-infected individuals: proposed clinical case definitions. *The Lancet Infectious Diseases* **10**, 791-802, doi:[https://doi.org/10.1016/S1473-3099\(10\)70170-5](https://doi.org/10.1016/S1473-3099(10)70170-5) (2010).
- 5 Thambuchetty, N. *et al.* The Epidemiology of IRIS in Southern India: An Observational Cohort Study. *Journal of the International Association of Providers of AIDS Care* **16**, 475-480, doi:10.1177/2325957417702485 (2017).
- 6 Vlasova-St Louis, I., Chang, C. C., Shahid, S., French, M. A. & Bohjanen, P. R. Transcriptomic Predictors of Paradoxical Cryptococcosis-Associated Immune Reconstitution Inflammatory Syndrome. *Open Forum Infect Dis* **5**, ofy157, doi:10.1093/ofid/ofy157 (2018).
- 7 Kannambath, S. *et al.* Genome-Wide Association Study Identifies Novel Colony Stimulating Factor 1 Locus Conferring Susceptibility to Cryptococcosis in Human Immunodeficiency Virus-Infected South Africans. *Open Forum Infect Dis* **7**, ofaa489, doi:10.1093/ofid/ofaa489 (2020).
- 8 Khaw, Y. M. *et al.* Th1-Dependent Cryptococcus-Associated Immune Reconstitution Inflammatory Syndrome Model With Brain Damage. *Front Immunol* **11**, 529219, doi:10.3389/fimmu.2020.529219 (2020).

REVIEWERS' COMMENTS

Reviewer #1 (Remarks to the Author):

It is a well-conducted study that investigates the pathogenic role of T cell/neuron interactions in a C-IRIS model, which has significant potential to advance our understanding of C-IRIS pathogenesis. The authors addressed all comments from the previous review by providing new data and editing the text. I particularly appreciate the expansion of the lung function studies which solidified the conclusions linking the T-cell-induced damage of brain neurons and the observed respiratory problems. The presented interception of CCR5-mediated recruitment of CD4 T-cells to provide mechanistic proof and introduce a therapeutic approach represent a significant advance and possibly a future therapeutic intervention for C-IRIS patients.

Reviewer #2 (Remarks to the Author):

Kawano et al, R1

The authors responded to many of the comments and are to be applauded for their extensive work which could have applications to both the general fields of immunology as well as human disease. However, many of the criticisms were either not responded to or incompletely responded to. The use of vague relationships between mouse models and human disease suggests relationships between the two that are just not there. Clarifying these disparities will not take away from the findings, but will rather result in a better context of the findings and future research. Inclusion of a limitations paragraph as is done increasingly today in both clinical and basic science journals should also be undertaken as a convenient method of highlighting these differences. Specifics are as below:

L62: The authors note that cIRIS is associated with pulmonary nodules and pleural effusions but this does not result in hypoventilation as the mouse model described. Typically, patients will develop respiratory failure at the point of brain/uncal herniation, which does not mimic that of the mouse model. Salient reviews of cIRIS should be referenced and separated from descriptions of selected case reports that may distort the picture if quoted alone. In addition, the authors are really modeling what is called unmasking (without immune reconstitution) rather than cIRIS, which should be clarified. These points should be explicitly stated.

L105: Not of the references describe C-IRIS presenting with respiratory symptoms. #3 is a case of pulm cryptococcosis without c-IRIS (crypto can manifest as pulmonary disease alone which has little mortality without meningitis) Ref #4 and #6 do not provide the respiratory symptoms that would support the strength of the mouse model resembling that of human cIRIS.

L309: The described model only considers T cell function in generation of an inflammatory state. This does not reduce the significance of the model but should be explicitly stated in the discussion.

Statistics: Data should be compared using not total technical replicates but using an N that corresponds to the number of mice representing independent experiments using the mean/median from the technical replicates for each mouse. Figure legends should also specifically describe numbers of mice in a manner typical of other Nature journals, for example: Error bars indicate s.d., n=3 independent experiments. Student's t-test, *P <0.05, **P <0.01 and ***P <0.001. It is difficult to imagine that significant results could be generated with only 2 mice. If experiments utilized only two mice, an additional experiment should be conducted.

Response to the editor and the referees

A point-by-point response to Referee #1 comments:

- 1- It is a well-conducted study that investigates the pathogenic role of T cell/neuron interactions in a C-IRIS model, which has significant potential to advance our understanding of C-IRIS pathogenesis. The authors addressed all comments from the previous review by providing new data and editing the text. I particularly appreciate the expansion of the lung function studies, which solidified the conclusions linking the T-cell-induced damage of brain neurons and the observed respiratory problems. The presented interception of CCR5-mediated recruitment of CD4 T-cells to provide mechanistic proof and introduce a therapeutic approach represent a significant advance and possibly a future therapeutic intervention for C-IRIS patients

Response: We sincerely appreciate the comments and the constructive suggestions provided by reviewer #1 during the revision process. Your comments greatly contributed to the quality of our work.

A point-by-point response to Referee #2 comments:

- 1- The authors responded to many of the comments and are to be applauded for their extensive work which could have applications to both the general fields of immunology as well as human disease. However, many criticisms were either not responded to or incompletely responded to. The use of vague relationships between mouse models and human disease suggests relationships between the two that are just not there. Clarifying these disparities will not take away from the findings, but will rather result in a better context of the findings and future research. Inclusion of a limitations paragraph as is done increasingly today in both clinical and basic science journals should also be undertaken as a convenient method of highlighting these differences. Specifics are as below:

Response: We would like to take this opportunity to express our sincere appreciation to reviewer #2 for providing comments and constructive suggestions during the revision process. Accordingly, we added the following paragraph to the discussion “Although our mouse model mimics human C-IRIS, the inherent differences in biology and immune responses between the mouse C-IRIS model and human C-IRIS situation should be considered. For instance, C-IRIS patients develop respiratory failure at the point of brain/uncal herniation[1]. However, although our C-IRIS model mice show brain edema [2], which may trigger brain herniation, such mice do not show brain/uncal herniation when mice develop respiratory failure. The basis for these differences is not understood and merits further investigation. It is unlikely that any single animal model will fully recapitulate the human disease in all its details. Thus, developing other C-IRIS animal models that overcome current limitations or mimic paradoxical C-IRIS is essential. Further investigation of IRIS pathogenesis using our and future models would lead to developing therapies for C-IRIS”. Page 17, Lines 371-380.

- 2- L62: The authors note that cIRIS is associated with pulmonary nodules and pleural effusions but this does not result in hypoventilation as the mouse model described. Typically, patients will develop respiratory failure at the point of brain/uncal herniation, which does not mimic that of the mouse model. Salient reviews of cIRIS should be referenced and separated from descriptions of selected case reports that may distort the picture if quoted alone. In addition, the authors are really modeling what is called unmasking (without immune reconstitution) rather than cIRIS, which should be clarified. These points should be explicitly stated.

Response: We thank the reviewers for pointing out the main cause of hypoventilation in clinical C-IRIS. Accordingly, we added a paragraph about this critical point to the discussion

section in the revised manuscript: “Although our mouse model mimics human C-IRIS, the inherent differences in biology and immune responses between mice and humans should be considered. For instance, C-IRIS patients develop respiratory failure at the point of brain/uncal herniation[1]. However, although our C-IRIS model mice show brain edema[2], which may trigger brain herniation, such mice do not show brain herniation when mice develop respiratory failure. Page 17, Lines 371-380. In addition, we added and modified the introduction section. We also added “unmasking” to the introduction and discussion.

- 3- L105: Not of the references describe C-IRIS presenting with respiratory symptoms. #3 is a case of pulm cryptococcosis without c-IRIS (crypto can manifest as pulmonary disease alone which has little mortality without meningitis) Ref #4 and #6 do not provide the respiratory symptoms that would support the strength of the mouse model resembling that of human cIRIS.

Response: We thank the reviewer for raising this issue. We have replaced the needed references [1, 3, 4]. Page 5 line 106.

- 4- L309: The described model only considers T cell function in generation of an inflammatory state. This does not reduce the significance of the model but should be explicitly stated in the discussion.

Response: We thank the reviewer for the suggestion. We added a paragraph, “Because our model is a T cell-driven unmasking C-IRIS disease, we investigated T cell function in generating an inflammatory state in the present study.” to the discussion. Page 14, Lines 311-313.

- 5- Statistics: Data should be compared using not total technical replicates but using an N that corresponds to the number of mice representing independent experiments using the mean/median from the technical replicates for each mouse.

Response: In all studie instead of cell culture studies, “N” indicates the number of mice representing independent experiments. In cell cuture studies, “N’ indicates the number of biological samples. We indicated them in the legends.

- 6- Figure legends should also specifically describe numbers of mice in a manner typical of other Nature journals, for example: Error bars indicate s.d., n=3 independent experiments. Student’s t-test,*P <0.05, **P <0.01 and ***P <0.001. It is difficult to

imagine that significant results could be generated with only 2 mice. If experiments utilized only two mice, an additional experiment should be conducted.

Response: We appreciate the suggestion. We modified figure legends following nature communications "Guide to authors". In this study, our sample sizes are $n \geq 4$.

References

1. Guevara, N., et al., *A case report of a brain herniation secondary to cryptococcal meningitis with elevated intracranial pressure in a patient with Human Immunodeficiency Virus/Acquired immunodeficiency syndrome (HIV/AIDS)*. IDCases, 2022. **29**: p. e01554.
2. Khaw, Y.M., et al., *Th1-Dependent Cryptococcus-Associated Immune Reconstitution Inflammatory Syndrome Model With Brain Damage*. Front Immunol, 2020. **11**: p. 529219.
3. Lortholary, O., et al., *Incidence and risk factors of immune reconstitution inflammatory syndrome complicating HIV-associated cryptococcosis in France*. Aids, 2005. **19**(10): p. 1043-9.
4. Hu, Z., et al., *Pulmonary cryptococcal immune reconstitution syndrome in a person living with HIV: a case report*. Int J STD AIDS, 2020. **31**(3): p. 280-284.